# Intrinsically disordered intracellular domains control key features of the mechanically-gated ion channel PIEZO2

Clement Verkest[1,5], Irina Schaefer[2,5], Timo A. Nees[2,3], Na Wang [2], Juri M. Jegelka[2], Francisco J. Taberner[2,4] & Stefan G. Lechner [1,2✉]

A central question in mechanobiology is how mechanical forces acting in or on cells are transmitted to mechanically-gated PIEZO channels that convert these forces into biochemical signals. Here we examined the role of the intracellular domains of PIEZO2, which account for 25% of the channel, and demonstrate that these domains fine-tune properties such as poking and stretch-sensitivity, velocity coding and single channel conductance. Moreover, we show that the intrinsically disordered linker between the transmembrane helices twelve and thirteen (IDR5) is required for the activation of PIEZO2 by cytoskeleton-transmitted forces. The deletion of IDR5 abolishes PIEZO2-mediated inhibition of neurite outgrowth, while it only partially affected its sensitivity to cell indentation and does not alter its stretch sensitivity. Thus, we propose that PIEZO2 is a polymodal mechanosensor that detects different types of mechanical stimuli via different force transmission pathways, which highlights the importance of utilizing multiple complementary assays when investigating PIEZO function.

[1] Department of Anesthesiology, University Medical Center Hamburg-Eppendorf, Martinistrasse 52, 20246 Hamburg, Germany. [2] Institute of Pharmacology, Heidelberg University, Im Neuenheimer Feld 366, 69120 Heidelberg, Germany. [3] Clinic for Orthopedics and Trauma Surgery, Center for Orthopedics, Trauma Surgery and Spinal Cord Injury, Heidelberg University Hospital, Schlierbacher Landstrasse 200a, 69118 Heidelberg, Germany. [4] Present address: Instituto de Neurosciencias de Alicante, Universidad Miguel Hernández – CSIC, Alicante, Spain. [5] These authors contributed equally: Clement Verkest, Irina Schaefer. ✉email: s.lechner@uke.de

Virtually all cells of our organism are constantly exposed to mechanical forces of one kind or another. Thus, besides the ubiquitous gravitational and osmotic forces, some cells experience compression and stretch induced by externally applied mechanical stimuli or by organ distension, respectively, while others are exposed to shear stress exerted by circulating body fluids. Moreover, many cells generate traction forces acting on their surface when they explore their local environment. Accordingly, most cells are equipped with sensors that enable them to detect and convert mechanical stimuli into biochemical signals – a process called mechanotransduction – that trigger adaptive processes required to maintain cell, tissue and not least body integrity in an ever-changing mechanical environment.

Since their discovery in 2010[1], the mechanically activated ion channels PIEZO1 and PIEZO2 were shown to be of crucial importance for mechanotransduction in a variety of tissues. PIEZO1, for example, detects shear stress and stretch in erythrocytes, vascular endothelial cells, chondrocytes and bladder urothelial cells[2,3]. PIEZO2, on the other hand, appears to be particularly important for mechanotransduction in primary sensory afferents as it was shown to be involved in the detection of light touch, mechanical pain, proprioception, airway stretch and bladder distension[4–11]. Moreover, PIEZOs contribute to the regulation of processes such as neurite outgrowth[12], wound healing[13] and tumor cell dissemination[14], probably by detecting cell-generated traction forces acting on the plasma membrane during neurite extension and cell migration[15–18]. High-resolution cryo-EM studies showed that PIEZO1 and PIEZO2 oligomerize as homotrimers with a propeller-shaped quaternary structure[19–22] and subsequent structure-guided studies have revealed the role of several structural domains as well as single amino acids (AA) and interdomain interactions that control important functional properties such as channel inactivation, ion permeation, voltage gating and sensitivity to PIEZO modulating agents[23–33].

Despite our growing knowledge about the physiological roles of PIEZOs and our profound understanding of the structure-function relationship of PIEZOs, however, a fundamental question that remains open is how mechanical forces that act on the cell surface are transmitted to the channel in the first place. Two paradigms are commonly used to explain mechanical force-induced channel activation. The force-from-lipids model proposes that mechanically-induced membrane tension causes changes in the transbilayer pressure profile asymmetry, which leads to conformational changes and thus activation of PIEZOs[34–36]. The force-from-filament model, on the other hand, proposes that PIEZOs – like other mechanically-gated ion channels (e.g., NOMPC, TMC1)[37,38] – are tethered to the cytoskeleton, such that mechanically-induced movements of the cytoskeleton activate the channel by pulling or pushing it open from the intracellular side. Whether the two force transmission pathways are mutually exclusive or whether they act synergistically to activate PIEZOs, is, however, still unclear.

PIEZO2 comprises seven large intracellular domains that together account for ~25% of its AA sequence and that have – owing to the lack of structural information – been ignored by previous studies. Interestingly, these intracellular domains account for the majority of the size and AA sequence differences between PIEZO2 and PIEZO1, suggesting that they might, at least partially, determine the functional differences between the two channels. Moreover, some of the intracellular domains are partly encoded by alternatively spiced exons[39], which highlights a possible role in fine-tuning PIEZO2 splice variant function. Finally, considering their size and localization, the intracellular domains are ideally suited to mediate possible interactions between the cytoskeleton and the channel that might be involved PIEZO2 gating.

In this work we generate PIEZO2 mutants that lack the various intracellular domains and examine their role using electrophysiological recordings, TIRF microscopy and neurite outgrowth assays. The key finding of our study is that the intrinsically disordered linker between the transmembrane helices twelve and thirteen (IDR5) is required for the activation of PIEZO2 by cytoskeleton-transmitted forces, but appears to be dispensable for PIEZO2 activation by pressure-induced membrane stretch. Thus, our data suggests that PIEZO2 is a polymodal mechanosensor that detects different types of mechanical stimuli via different force transmission pathways, which highlights the importance of utilizing multiple complementary assays when investigating PIEZO function.

## Results

**Membrane indentation and pressure-induced membrane stretch activate PIEZO2 via different mechanisms.** According to three commonly used intrinsic disorder prediction algorithms, IUpred-L, PONDR-VSL2b and ESpritz-N[40–42], all seven intracellular domains of PIEZO2 have very high degrees of intrinsic disorder (Fig. 1a, b), that is they do not assume a well-defined tertiary structure. We thus hereafter refer to the intracellular domains as intrinsically disordered domains 1-7 (IDR1-7). To examine the role of the IDRs we generated PIEZO2 mutants that lack individual IDRs (IDR1del – IDR7del, Supplementary Fig. 1) and assessed their function in Neuro2a-PIEZO1-KO cells, which completely lack endogenous mechanotransduction currents[25]. We first compared PIEZO2 with IDR1del–IDR7del-mediated currents evoked by mechanical indentation of the cell membrane using whole-cell patch-clamp recordings (Fig. 1c) – a commonly used approach for studying PIEZO2 function. Strikingly, IDR5del-mediated currents exhibited dramatically reduced amplitudes (IDR5: $0.34 \pm 0.07$ nA vs. PIEZO2 $1.13 \pm 0.12$ nA at $5.2$ μm indentation, Fig. 1c, d) and slightly, yet significantly, increased activation thresholds (Supplementary Fig. 3a). The inactivation kinetics and the reversal potential were not affected by the deletion of IDR5 (Fig. 1e and Supplementary Fig. 3b, c). Moreover, IDR3del-mediated currents were more than twice as big as full-length PIEZO2-mediated currents (IDR3del: $2.6 \pm 0.43$ nA vs. PIEZO2 $1.13 \pm 0.12$ nA at $5.2$ μm Fig. 1c, d), while exhibiting similar activation thresholds, inactivation kinetics and reversal potentials (Fig. 1e and Supplementary Fig. 3a–c). Interestingly, all other IDR-deletions produced poking-evoked currents that were indistinguishable from full-length PIEZO2 currents (Fig. 1d, e, Supplementary Fig. 3). Consistent with the observation that all IDR-deletions appeared to be functional, immunocytochemistry showed that all seven mutants are properly trafficked to the plasma membrane (Supplementary Fig. 2).

A peculiar feature of PIEZO2 is that its current amplitudes do not only increase as a function of stimulation amplitude, but also get bigger as the stimulation velocity increases[27,43]. Hence, we asked if the observed differences in the displacement-response functions (Fig. 1d) resulted from differences in the velocity sensitivity. To this end we compared currents evoked by ramp-and-hold stimuli with a ramp speed of 1 μm/ms with currents evoked by stimuli with a ramp speed of 0.25 μm/ms. As previously described[27], full-length PIEZO2 current amplitudes were ~60% smaller when evoked with slow mechanical stimuli (0.25 μm/ms) as compared to fast stimuli (1 μm/ms, Fig. 1f and g). Similar velocity dependences were observed for all other IDR deletions, except for IDR2del, which produced significantly smaller currents at a stimulation velocity of 0.25 μm/ms (Fig. 1f, g).

Another commonly used experimental approach for studying PIEZO channel function is the so-called pressure-clamp technique[44]. Here, the currents are recorded in the cell-attached mode of the patch-clamp technique and the channels are activated by stretching the membrane inside the patch-pipette

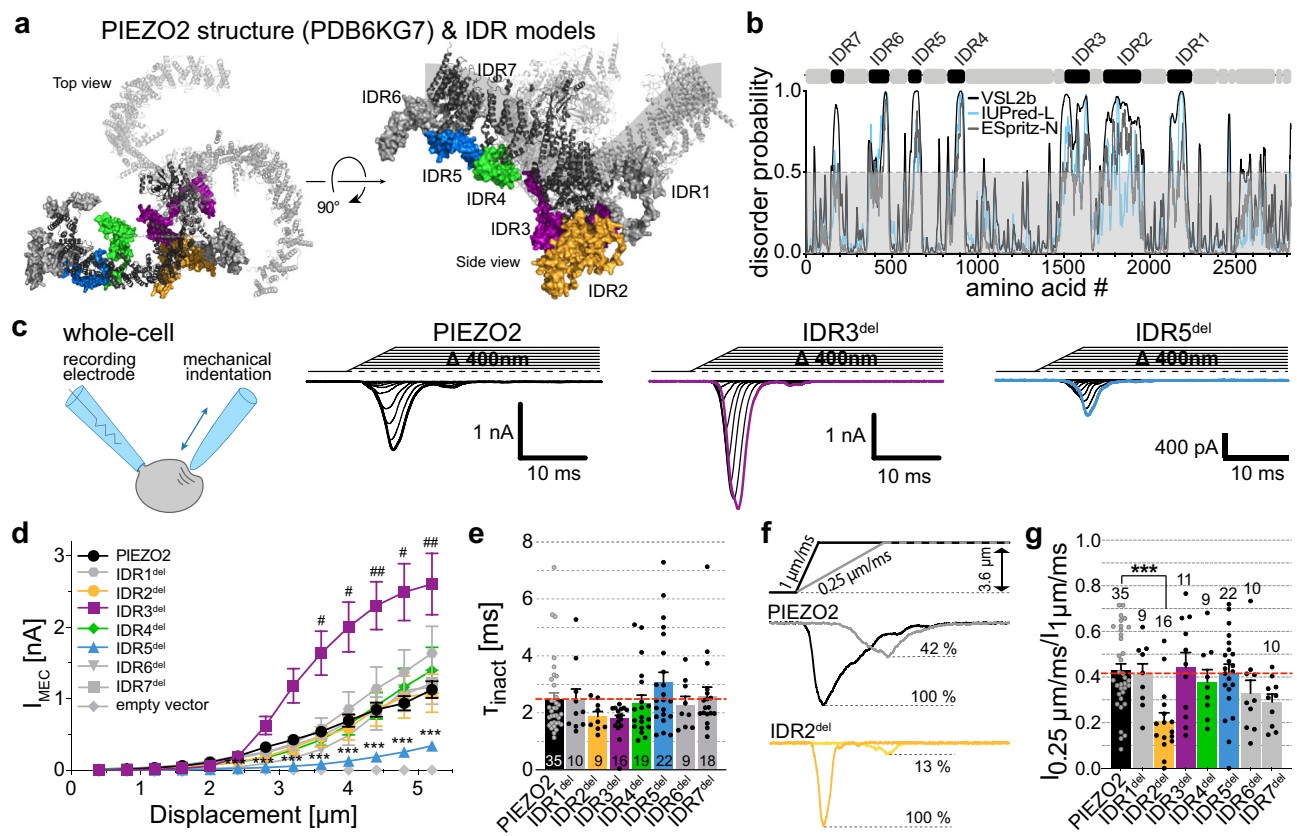

**Fig. 1 Deletion of IDR2, IDR3 and IDR5 alters membrane indentation-evoked PIEZO2 currents. a** Top and side view of the mouse PIEZO2 structure. IDR regions were modeled with SWISS-MODEL and combined with the partial cryo-EM PIEZO2 structure (6KG7). Note, QMEAN-values of the IDRs were low and thus the model solely serves as an orientation guide for the approximate size and localization of the IDRs. **b** mPIEZO2 amino acid sequence-based disorder predictions (bottom) determined using IUPRED-L (blue), PONDR-VSL2b (black) and ESpritz-N (gray). Positions of the IDRs are indicated above the graph. **c** Recording configuration cartoon and example traces of whole-cell currents evoked by mechanical indentation of N2a cells expressing PIEZO2 (black, left), IDR3$^{del}$ (purple, middle) and IDR5$^{del}$ (blue, right). **d** Displacement-responses curves of peak current amplitudes of PIEZO2, IDR$^{del}$ mutants and empty vector in N2a-P1KO cells. Symbols are mean ± s.e.m. N-numbers vary within groups because large mechanical stimuli occasionally damage the cells; PIEZO2 ($N_{0.4-4\,\mu m} = 35$, $N_{4.4\,\mu m} = 33$, $N_{4.8\,\mu m} = 30$, $N_{5.2\,\mu m} = 29$), IDR1$^{del}$ ($N = 10$), IDR2$^{del}$ ($N_{0.4-4\,\mu m} = 9$, $N_{4.4\,\mu m} = 8$, $N_{4.8\,\mu m} = 6$, $N_{5.2\,\mu m} = 5$), IDR3$^{del}$ ($N_{0.4-4.4\,\mu m} = 16$, $N_{4.8\,\mu m} = 14$, $N_{5.2\,\mu m} = 13$), IDR4$^{del}$ ($N_{0.4-4.4\,\mu m} = 19$, $N_{4.8\,\mu m} = 17$, $N_{5.2\,\mu m} = 16$), IDR5$^{del}$ ($N = 22$), IDR6$^{del}$ ($N_{0.4-4.8\,\mu m} = 9$, $N_{5.2\,\mu m} = 6$), IDR7$^{del}$ ($N_{0.4-2.8\,\mu m} = 18$, $N_{3.2\,\mu m} = 16$, $N_{3.6-4.0\,\mu m} = 15$ $N_{4.4\,\mu m} = 13$, $N_{4.8\,\mu m} = 11$, $N_{5.2\,\mu m} = 7$), empty vector ($N = 15$). Comparison with Kruskal–Wallis test and Dunn's post-test. *P*-values for PIEZO2 vs IDR5$^{del}$ (from left to right): ***0.0000007, ***0.0000003, ***0.0000002, ***0.000001, ***0.000002, ***0.00002, ***0.000138, ***0.000041. *P*-values for PIEZO2 vs IDR3$^{del}$ (from left to right): #0.0239, #0.0232, ##0.0054, #0.0186, ##0.0098. **e** Comparison of the inactivation time constants ($\tau_{inact}$) of PIEZO2 and IDR$^{del}$-mutants using with Kruskal–Wallis ($p = 0.1531$). N-numbers are indicated within bars. **f** Example traces of PIEZO2 and IDR2$^{del}$ currents evoked by stimuli of different velocities and (**g**) comparison of the current amplitude ratios ($I_{0.25\mu m/ms}/I_{1\mu m/ms}$) of PIEZO2 and all IDR-deletions using One-Way ANOVA, $F(7, 114) = 3.771$ and Dunnett's post-test (PIEZO2 vs IDR2$^{del}$, ***$p = 0.0002$). Bars represent means ± s.e.m with ratios values from individual cells shown as black circles and N-numbers per group being indicated above the bars. Source data are provided as a Source Data file.

by application of negative pressure (Fig. 2a). Consistent with previous reports[22,25,45,46], we observed stretch-evoked PIEZO2 currents in ~20% (8/38) of the recorded cells (Fig. 2a, b). Interestingly, the proportion of cells that exhibited pressure-induced currents was more than twice as big amongst cells expressing IDR1$^{del}$ (9/19 cells), IDR2$^{del}$ (8/18 cells), IDR4$^{del}$ (25/42 cells) and IDR5$^{del}$ (10/22 cells), but this difference was only statistically significant for IDR4$^{del}$ (Fig. 2a, b). Since the pressure-evoked PIEZO2 currents never saturated in the pressure range in which the recordings were stable (pressures below –80 mmHg frequently ruptured the membrane) and, moreover, did not exhibit clearly discriminable single-channel openings or peaks at higher pressures (Fig. 2a, Supplementary Fig. 4), we were unable to estimate the total number of channels in the patch and thus could not calculate dwell times and open probabilities. Hence, in order to statistically compare the pressure-evoked currents, we determined the total charge transfer by measuring the area under

the curve (AUC) over the time of the pressure stimulus. This analysis revealed, that IDR4$^{del}$ did not only respond more frequently but also generated significantly larger currents in response to membrane stretch, (PIEZO2: 1.11 ± 0.41 pC vs. IDR4: 4.05 ± 1 pC at −60 mmHg, Fig. 2c). The analysis of single-channel openings at low pressures and varying holding potentials, showed that IDR3$^{del}$ has a significantly larger single-channel conductance than full-length PIEZO2 (IDR3$^{del}$, 34.07 ± 1.26 pS vs. PIEZO2, 23.4 ± 1.14 pS, mean ± SEM), whereas all other IDR-deletions had single-channel conductances that were in the same range (Fig. 2d–f).

Taken together, our data suggests that IDR3 is involved in determining the single-channel conductance of PIEZO2, whereas IDR2 appears to control its velocity sensitivity. The most intriguing observation, however, was that deletion of IDR4 increased stretch-sensitivity but did not affect poking-sensitivity, while the deletion of IDR5 and IDR2 (at slow stimulation

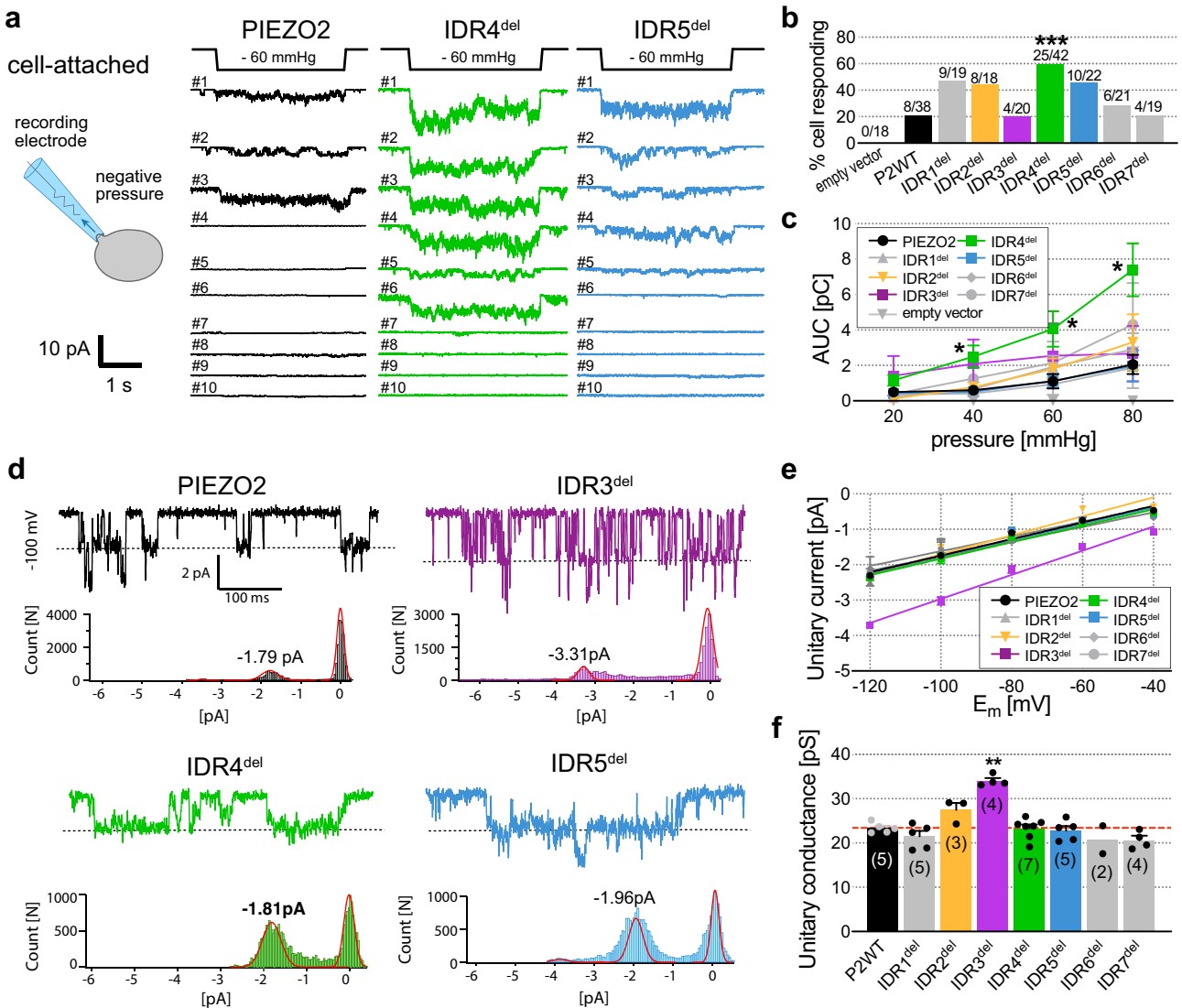

**Fig. 2 Deletion of IDR3 and IDR4 impairs stretch- and single channel-responses of PIEZO2. a** Cartoon illustrating the pressure-clamp technique (left) and example traces (right) of stretch-activated PIEZO2, IDR4$^{del}$ and IDR5$^{del}$-currents recorded from 10 different cells in cell-attached mode (holding potential = −100 mV, pressure stimulus = −60 mmHg). **b** Comparison of the proportion of cells responding to pressure-induced membrane stretch amongst cells expressing GFP (empty vector), PIEZO2 and the individual IDR$^{del}$ mutants. Labels above bars (X/Y) indicate how many (X) from Y tested cells responded. Fisher's exact test, PIEZO2 vs IDR1$^{del}$ ns $p = 0.0646$, PIEZO2 vs IDR4$^{del}$ *** $p = 0.0006$, PIEZO2 vs IDR5$^{del}$ ns $p = 0.07801$. **c** Stimulus (pressure)–response (area under the curve, AUC) curves of PIEZO2 and IDR$^{del}$ mutants (for example traces see Supplementary Fig. 4). Symbols represent means ± s.e.m. Comparison with Kruskal–Wallis test ($p = 0.0569$ −20 mmHg, $p = 0.0222$ −40 mmHg, $p = 0.0041$ −60 mmHg, $p = 0.0043$ −80 mmHg) and Dunn's post-test, PIEZO2 vs IDR4 (*$p = 0.0427$ −40 mmHg, *$p = 0.0427$ −60 mmHg, *$p = 0.0483$ −80 mmHg). N-numbers of cells per group are the same as in **b**. **d** Example traces (top) of stretch-activated PIEZO2 (black), IDR3$^{del}$ (purple), IDR4$^{del}$ (green) and IDR5$^{del}$-currents (blue) are shown together with corresponding current amplitude distribution histograms (bottom, peak values of Gaussian fits – i.e., unitary currents – are indicated). **e** Linear regression fits of the I-V plots (mean ± s.e.m. unitary currents vs. holding potential) of PIEZO2 and the indicated IDR$^{del}$-mutants. N-numbers are the same as in **f. f** Unitary conductance of PIEZO2 and IDR$^{del}$ mutants. Bars represent means ± s.e.m., with individual values from each cell shown as black circles and N-numbers shown in brackets. Comparison with Kruskal–Wallis ($p = 0.0069$) and Dunn's post-test, PIEZO2 vs IDR3$^{del}$, *$p = 0.0096$. Source data are provided as a Source Data file.

velocities) had the opposite effect – i.e., reduced poking-sensitivity but normal stretch-sensitivity – which suggests that poking and stretch activate PIEZO2 via two different mechanisms that involve different channel domains.

**IDR2 fine-tunes velocity sensitivity of PIEZO2.** IDR2 connects the clasp domain with transmembrane helical unit 8 (THU8) and is the longest intracellular domain of PIEZO2 (239 AAs, Supplementary Fig. 1)[22]. Since, the equivalent domain in PIEZO1 is

much shorter (99 AAs) and has less than 20% sequence similarity with PIEZO2-IDR2, we hypothesized that PIEZO1 might have a similar velocity sensitivity as PIEZO2-IDR2$^{del}$. To test this hypothesis, we compared the current amplitudes of PIEZO2, IDR2$^{del}$ and PIEZO1 evoked by a series of mechanical ramp-and-hold stimuli with increasing ramp speeds (Fig. 3a, b). These recordings revealed that PIEZO1 is significantly less sensitive to slow mechanical stimuli than PIEZO2 and exhibits a velocity dependence that is similar to that of IDR2$^{del}$, suggesting that IDR2 confers sensitivity to slow mechanical stimuli to PIEZO2.

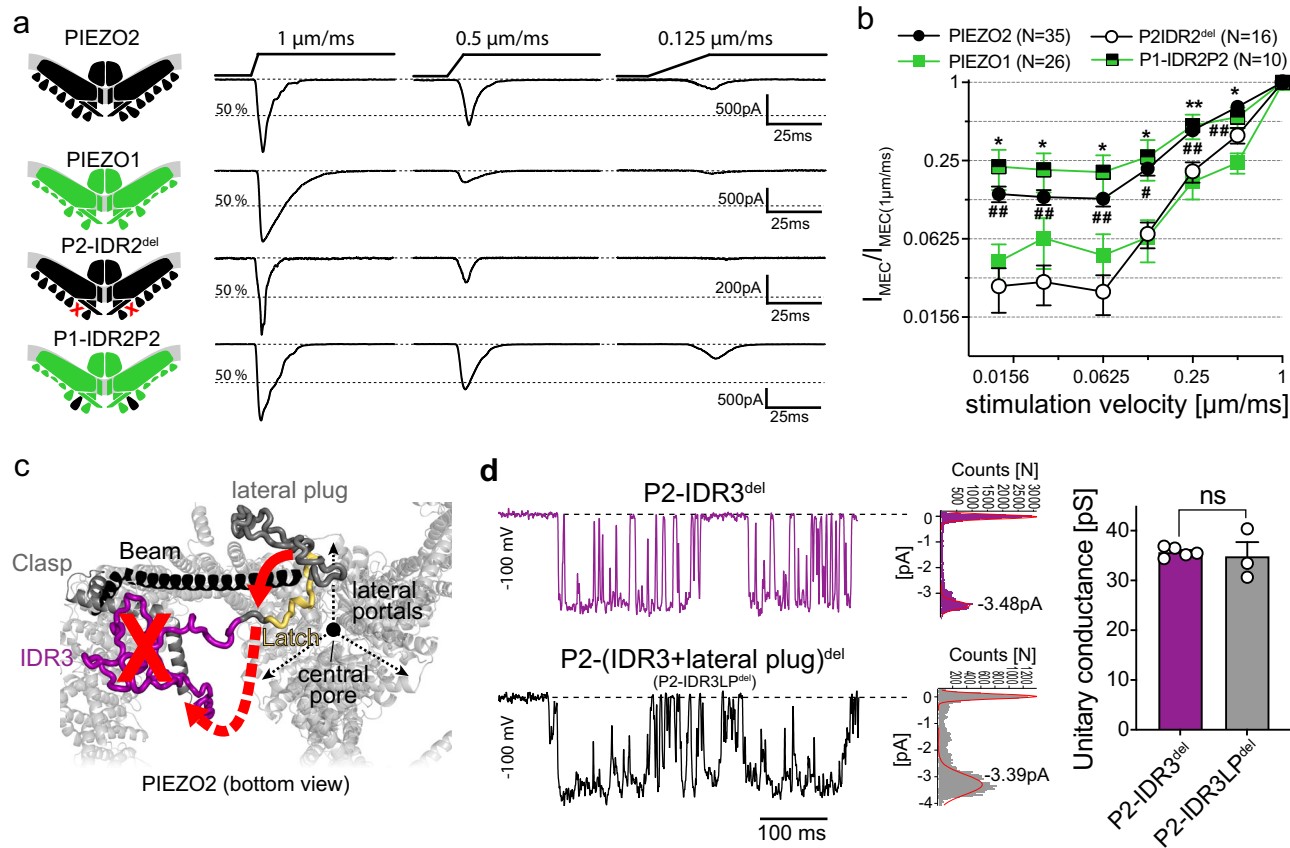

**Fig. 3 IDR2 and IDR3 control velocity sensitivity and single-channel conductance of PIEZO2, respectively. a** Example traces of currents evoked by mechanical ramp-and-hold stimuli with the indicated ramp speeds (right) in cells expressing the illustrated PIEZO channels/chimeras (left). **b** Comparison of the velocity sensitivities of PIEZO2 ($N = 35$), P2-IDR2del ($N = 16$), PIEZO1 ($N = 26$) and P1-IDR2P2 ($N = 10$). Graph shows the mean normalized current amplitudes ±s.e.m. of the indicated channels, as a function of stimulation velocity. Comparison with Kruskal–Wallis and Dunn's post-test was performed separately for each tested velocity: $P$-values for PIEZO2 vs IDR2del (from left to right): ##0.0012, ##0.0036, ##0.0015, #0.0237, ##0.005 ##0.0047 and $P$-Values for PIEZO1 vs P1-IDR2P2 (from left to right): *0.0196, *0.025, *0.0105, *0.0379, **0.0076, *0.0219. **c** Cartoon illustrating the possible structural changes induced by IDR3 deletion. Deletion of IDR3 (red X) might cause dislocation (red dashed arrow) of the latch (yellow) and the lateral plug (red arrow) such that the lateral portals become unblocked and single-channel conductance increases. **d** Example traces of stretch-evoked single-channel currents mediated by IDR3del (top, purple) and IDR3del + LateralPlugdel mutant (bottom, black) are shown alongside with the corresponding current amplitude distribution histograms with Gaussian fits of the single channel opening peak. Bar graphs (right) represent mean ± s.e.m. unitary conductances of IDR3del and IDR3del + LateralPlugdel, with individual values from 5 (P2-IDR3del) and 3 (P2-IDR3LPdel) independent cells shown as white circles. Comparison with Mann–Whitney test ($p = 0.5714$). Source data are provided as a Source Data file.

To corroborate this hypothesis, we generated a PIEZO1 chimera in which we replaced the THU8-to-clasp linker with IDR2 of PIEZO2 (P1-IDR2P2). Strikingly, P1-IDR2P2 was significantly more sensitive to slow mechanical stimuli than PIEZO1 and exhibited the same velocity sensitivity as full-length PIEZO2 (Fig. 3a, b). Interestingly, neither the displacement-response curve of currents evoked by the standard stimulation velocity of 1 µm/ms, nor the mechanical threshold, the inactivation kinetics and the reversal potential, were affected by the insertion of P2-IDR2 into PIEZO1 (Supplementary Fig. 5a–e). Likewise, we did not observe any significant differences in the stretch sensitivity of PIEZO1 and P1-IDR2P2 in cell-attached recordings (Supplementary Fig. 5f–j). Hence, our results suggest a very specific role of IDR2 in fine-tuning velocity sensitivity of PIEZO channels and provide another example for a channel domain that is required for membrane indentation but not membrane stretch-induced activation of PIEZO channels.

**IDR3 is required for lateral plug function.** Our initial electrophysiological characterization showed that deletion of IDR3

increases single-channel conductance and, accordingly, also poking-evoked currents of PIEZO2 Figs. 1 and 2). IDR3 links the clasp domain to the latch domain, which on its other end is connected to the beam domain by a 44 AA long linker[22] (Fig. 3c, Supplementary Fig. 1). We had previously shown that this beam-to-latch linker controls ion permeation of PIEZO2 and PIEZO1[27] and Geng et al. later found that it does so by acting as a plug that blocks the lateral ion-conducting portals of PIEZO channels[26]. Considering that IDR3 spans a distance of approximately 7 nm, we hypothesized that deletion of IDR3 causes dislocation of the adjacent lateral plug, such that the lateral portals are unblocked and single-channel conductance is increased (Fig. 3c). To test this hypothesis, we generated a PIEZO2 mutant which in addition to IDR3 lacked the lateral plug (P2-(IDR3 + lateral plug)del). Single-channel recordings showed that the unitary conductance of IDR3del and P2-(IDR3 + lateral plug)del did not significantly differ (IDR3del: 35.7 ± 0.45 pS vs. P2-(IDR3 + lateral plug)del: 34.83 ± 2.90 pS; Fig. 3d), suggesting that the lateral portals are indeed unblocked in IDR3del such that additionally deleting the lateral plug does not further increase single-channel conductance. Hence, rather than being directly involved in controlling ion

permeation, IDR3 appears to be an important flexible structural element that ensures proper positioning of the lateral plug.

**Deletion of IDR5 does not alter PIEZO2 expression, clustering or mobility.** Finally, we considered the role of IDR5 in PIEZO2 function. A simple explanation for the small current amplitudes of IDR5[del] could be that the deletion of IDR5[del] causes a trafficking defect, which would result in less channels being inserted into the plasma membrane and hence smaller whole-cell currents, while still allowing single-channel recordings in the cell-attached mode. To enable a reliable quantification of PIEZO2 and IDR5[del] channel clusters in the plasma membrane, we generated PIEZO2 and IDR5[del] channels that were tagged with the red fluorescent protein mScarlet (Supplementary Fig. 6a). Both channels were indistinguishable from their untagged counterparts in patch-clamp recordings, indicating that the C-terminal mScarlet tag did not affect channel function (Supplementary Fig. 6b–j). Total internal reflection fluorescence (TIRF) microscopy showed that the number of channel clusters per $\mu m^2$ membrane at the cell-substrate interface was, in fact, even slightly higher in IDR5[del]-expressing cells than in cells expressing full-length PIEZO2 (Fig. 4a, b). Moreover, the cluster diameters, which were estimated by 2D gaussian fits (Fig. 4a), were similar for both channels (Fig. 4c). Previous live-cell TIRF studies had shown that PIEZO1 clusters are mobile in the membrane and suggested that altered clustering and/or diffusion might affect channel function[15,47]. We thus next asked if changes in lateral mobility account for the reduced mechanosensitivity of IDR5[del]. To this end we performed live-cell time-lapse TIRF imaging and tracked the movement of individual channel clusters using the TrackMate plugin of ImageJ[48] (Supplementary Movie S1 and S2). The tracks were then classified into different categories using the ImageJ TraJClassifier plugin, which distinguishes between tracks that result from (i) diffusion, (ii) sub-diffusion, (iii) confined movement and (iv) directed movement[49]. The great majority of the PIEZO2 clusters moved by diffusion and subdiffusion ($37.4 \pm 18.3\%$ and $52.8 \pm 16.2\%$), while a small proportion exhibited confined movement and directed movement. IDR5[del] clusters exhibited a very similar mobility pattern, with $37.9 \pm 10.9\%$ of the clusters moving by normal diffusion, $54.7 \pm 8.6\%$ by subdiffusion and small fractions of $5.1 \pm 3.9\%$ and $2.2 \pm 1.9\%$ showed confined and directed movement, respectively (Fig. 4d–f). Most importantly, the mean square displacements (MSD) and the diffusion rates of PIEZO2 and IDR5[del] clusters from the different movement categories were also identical (Fig. 4d, e and Supplementary Fig. 6k). Hence, the deletion of IDR5 neither affects the overall membrane expression level of PIEZO2 nor its lateral mobility within the membrane. To further corroborate these findings, we quantified the cell-surface expression levels of PIEZO2 and IDR5[del] using biotinylation and subsequent western blot analysis. Consistent with the quantification of the TIRF images (Fig. 4a–c), we found no differences in the membrane expression levels of IDR5[del] compared to full-length PIEZO2 with the biotinylation assay (Fig. 4g).

Another possible explanation for the functional deficit of IDR5[del] is that the deletion induces conformational changes in other domains that ultimately render the channel less sensitive to membrane indentation. To test this hypothesis, we generated PIEZO2 mutants in which the overall length of IDR5 was preserved, but AAs that could potentially be important for the function of IDR5 were mutated. IDR5, comprises a large proportion of negatively charged AAs (33/60, Fig. 4h and Supplementary Fig. 1), which prompted us to hypothesize that the negative charges might be essential for the function of IDR5. We thus generated PIEZO2 mutants in which short stretches of negatively charged AAs in IDR5 were substituted by uncharged glycine and poly-alanine stretches (Fig. 4h, PolyA1, PolyA3, PolyA4; PolyA2 cloning failed despite several attempts). While the currents mediated by PolyA3 and PolyA4 were indistinguishable from full-length PIEZO2 currents, PolyA1-mediated currents were significantly smaller and resembled those of IDR5[del] with respect to amplitudes and mechanical activation thresholds (Fig. 4i–l).

Taken together, the TIRF experiments and the biotinylation assays together with the observation that almost 50% of the IDR5[del]-expressing cells exhibited pressure-evoked currents (Fig. 2), strongly suggest that altered membrane trafficking, clustering or mobility do not account for the small poking-evoked current amplitudes of IDR5[del]. The fact that substituting the negatively charged AA stretch E625–E630 with uncharged alanines (PolyA1) was sufficient to reproduce the phenotype of IDR5[del], further suggests that IDR5 is not just a structural motif that is required for maintaining the overall tertiary structure of the channel, but is indeed directly involved in the activation of PIEZO2 by mechanical cell indentation.

**IDR5 is required for PIEZO2 activation by cytoskeleton-transmitted forces.** Previous studies that had investigated the effect of the actin cytoskeleton-disrupting drug Cytochalasin-D (Cyto-D) on mechanosensitivity, have shown that the activation of PIEZOs by mechanical indentation of the cell in whole-cell recordings requires an intact cytoskeleton whereas pressure-induced activation cell-attached mode does not[32,33,50–52]. In this context it should be kept in mind, that the membrane is attached to the cytoskeleton such that cytoskeleton disruption might also slightly affect membrane tension. However, considering the proposed requirement for an intact cytoskeleton as well as the possibility that PIEZO2 – like other mechanically-gated ion channels[37,38,53,54] – might be tethered to the cytoskeleton, together with the intracellular localization of IDR5 and our observation that deletion of IDR5 selectively reduces membrane indentation-induced but not pressure-evoked currents (Figs. 1d and 2), we hypothesized that IDR5 might be involved in force transmission from the cytoskeleton to the channel.

To test this hypothesis, we examined the effect of Cyto-D treatment on PIEZO2 and IDR5[del]-mediated currents. As previously shown for PIEZO1[32,50,52] and PIEZO2[33,51], treatment with Cyto-D significantly reduced poking-evoked current amplitudes of PIEZO2 (Fig. 5a, b) without affecting the mechanical activation threshold and inactivation kinetics (Supplementary Fig. 7a, b). By contrast, pressure-induced PIEZO2 currents in cell-attached recordings, tended to be larger and more frequent after Cyto-D treatment (Fig. 5e, g), which was consistent with previous reports describing similar effects of Cyto-D treatment on PIEZO1 currents[50,52]. Strikingly, Cyto-D treatment did not alter the amplitudes or other parameters of poking-evoked IDR5[del]-currents (Fig. 5a, b, Supplementary Fig. 7a, b), nor did it increase the proportion of cells with or the size of pressure-induced currents in cell-attached recordings (Fig. 5f, h). To further examine the role of the actin cytoskeleton in PIEZO2 activation, we also tested the effect of Jasplakinolide, which stabilizes actin filaments by inhibiting their disassembly and promoting the polymerization of actin monomers. Interestingly, Jasplakinolide treatment neither altered full-length PIEZO2 nor IDR5[del]-mediated poking-evoked whole-cell currents (Fig. 5c and Supplementary Fig. 7a, b). Finally, we tested if the inhibitory effect of Cyto-D treatment on poking-evoked PIEZO2 currents was specific for actin cytoskeleton disruption or if reducing the cell stiffness by other means has the same effect. To this end we examined the effect of Nocodazole treatment, which interferes

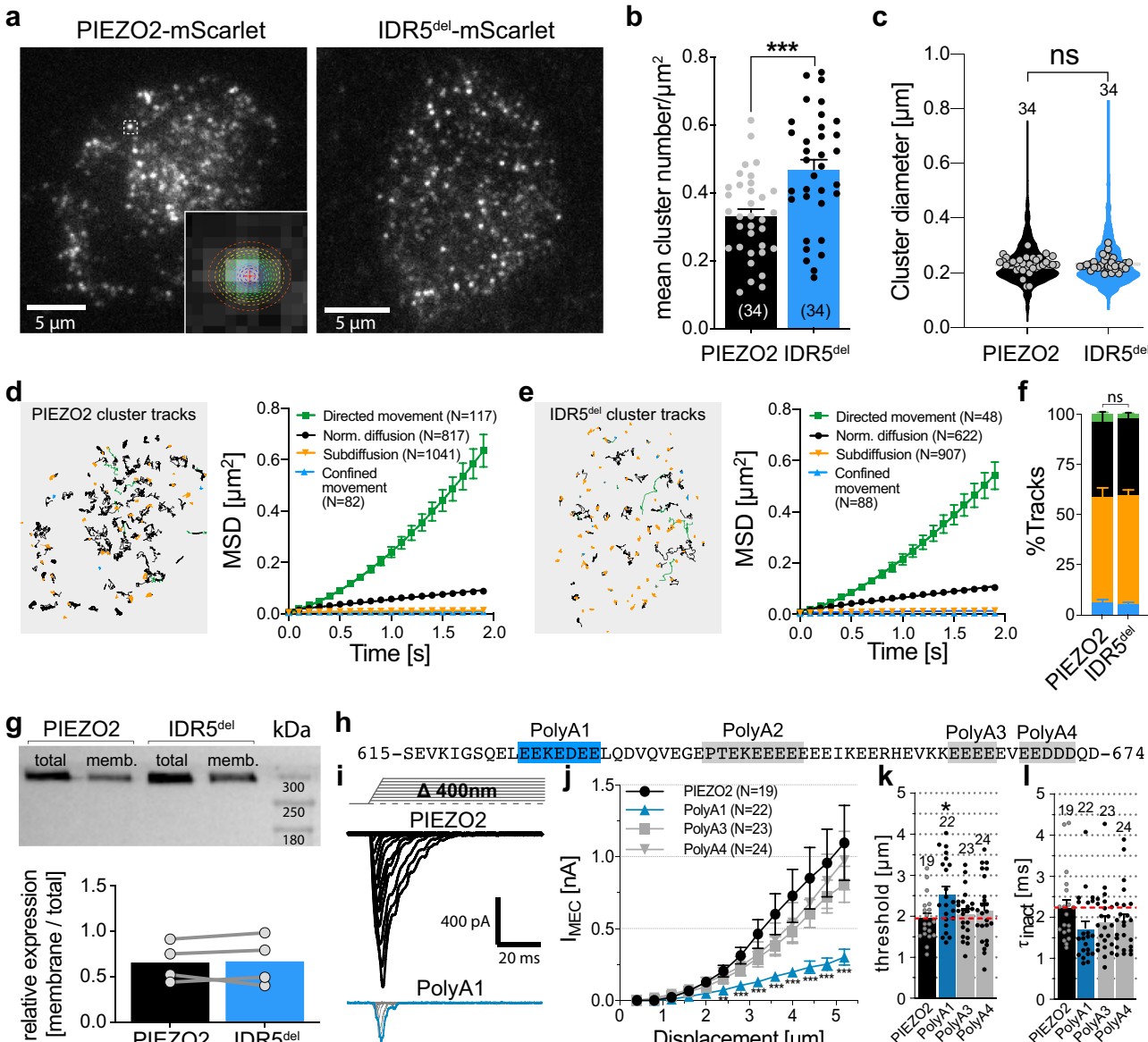

**Fig. 4 IDR5 deletion does not affect PIEZO2 trafficking, clustering or lateral diffusion in the plasma membrane. a** Example TIRF images of PIEZO2-mScarlet and IDR5del-mScarlet clusters. Inset shows Gaussian fit of a PIEZO2 cluster. **b** Mean cluster densities of 34 PIEZO2-mScarlet (black) and 34 IDR5del-mScarlet (blue) expressing cells were compared with unpaired *t*-test (***$p = 0.0003$). Error bars represent s.e.m. and means from individual cells are shown as circles. **c** Violin plot of individual cluster diameters (PIEZO2, $N = 2153$ and IDR5del, $N = 2765$) and mean cluster diameters per cell (gray circles, $N = 34$). Mean cluster diameters were compared using Mann–Whitney test ($p = 0.3152$). Tracks of the clusters shown in (**a**) (left) for PIEZO2 (**d**) and IDR5del (**e**) and plots of the Mean-Squared-Displacement as a function of lag time (right) for the 4 track categories. Symbols represent means ± s.e.m. Tracks from 2057 PIEZO2 and 1665 IDR5del clusters from 13 cells each were analyzed (N-Numbers for track-categories are provided in graph).
**f** Proportions of clusters assigned to the different track categories from 13 PIEZO2 and 13 IDR5del-expressing cells are shown in a stacked bar graph (mean ± s.e.m.) and compared using Mann–Whitney test, (Directed $p = 0.2555$, Norm. Diffusion $p = 0.9197$, Subdiffusion $p = 0.6597$, Confined $p = 0.99$; color-code same as in **d**). **g** Western blot of whole-cell lysate (total) and biotinylated membrane fraction (memb.) of PIEZO2 and IDR5del-transfected N2a-P1KO cells and bar graph of densitometric quantification (means with individual values from 4 independent assays. Mann–Whitney test, ($p = 0.7553$).
**h** PIEZO2-IDR5 sequence with positions of PolyA substitutions highlighted. **i** Example traces of poking-evoked PIEZO2 and PolyA1-mediated whole-cell currents. **j** Displacement-responses curves of current amplitudes of PIEZO2 and PolyA mutants. For each stimulus, current amplitudes were compared with Kruskal–Wallis test and Dunn's post-test. *P*-Values for PIEZO2 vs PolyA1 (from left to right): **0.004, ***0.0007, ***0.0003, ***0.0002, ***0.0001, ***0.0002, ***0.0002, ***0.0002). **k, l** comparisons of the mean ± s.e.m. mechanical activation thresholds and inactivation time constants of PIEZO2 and PolyA1 mutants with One-Way ANOVA, ($F_{(3, 84)} = 2.863$; Dunnett's post-test PIEZO2 vs PolyA1, *$p = 0.0211$) and Kruskal–Wallis test ($p = 0.1997$), respectively. N-numbers per group are provided above each bar. Source data are provided as a Source Data file.

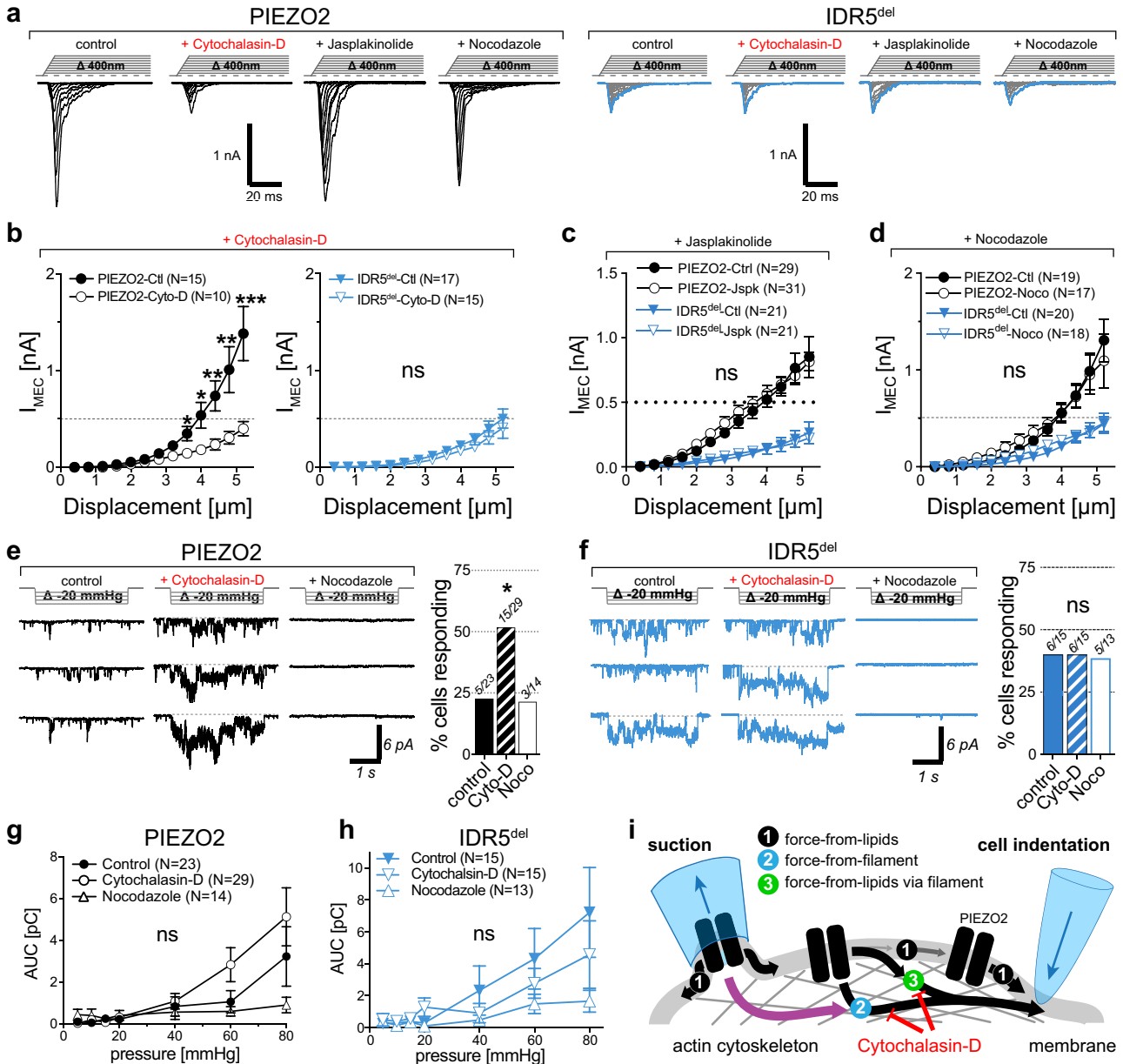

**Fig. 5 Actin cytoskeleton disruption impairs PIEZO2- but not IDR5$^{del}$-mediated membrane indentation-evoked currents. a** Example traces of poking-evoked PIEZO2 (left) and IDR5$^{del}$ (right) whole-cell currents recorded under the indicated conditions. **b** Displacement-responses curves of peak current amplitudes of PIEZO2 (left) and IDR5$^{del}$ (right) in the absence (control) or after treatment with Cytochalasin-D (15 min, 1 μM). Displacement-responses curves of peak current amplitudes of PIEZO2 and IDR5$^{del}$ in the absence (control) or in the presence of Nocodazole (30 min, 1 μM) (**d**) or Jasplakinolide (1 h, 200 nM) (**c**). For **b–d** symbols are mean ± s.e.m. Comparison of treated vs. untreated cells using Mann–Whitney test: *P*-Values for PIEZO2-Ctl vs PIEZO2-Cyto-D (from left to right): *0.047, *0.019, **0.009, **0.004, ***0.0009. N-numbers of cells per group are indicated in respective graph legend. Representative example traces of pressure-evoked currents recorded at -100mV in the cell-attached mode from untreated (control, left), Cytochalasin-D treated (middle) and nocodazole treated (right) N2a cells expressing PIEZO2 (**e**) and IDR5$^{del}$ (**f**) as well as comparison of the proportions of cells responding to stretch under the respective conditions. Fisher's exact test, PIEZO2-control vs Cytochalasin-D **p* = 0.0441. N-number of cells are indicated in the graph. Pressure-responses curves of PIEZO2 (**g**) and IDR5$^{del}$ (**h**) mediated pressure-evoked currents recorded in the indicated conditions. Symbols represent means ± s.e.m., which were compared using the Kruskal–Wallis test. N-number of cells per group is indicated in the graph legend. **i** Cartoon depicting the possible gating mechanisms of PIEZO2 in the pressure-clamp and the poking assays. Negative pressure applied to cell-attached patches (suction, left) stretches the membrane and supposedly activates PIEZOs by force-from-lipids. Poking of the cell surface with a fire-polished glass pipette inevitable causes both, stretch and cytoskeletal movements. Hence, in this configuration PIEZOs might be activated by (1) force-from-lipids, (2) by forces directly transmitted to channel by the cytoskeleton (force-from-filament), or (3) the cytoskeleton might relay forces to the membrane such that PIEZOs are eventually gated by membrane stretch (force-from-lipids-via-filament). Source data are provided as a Source Data file.

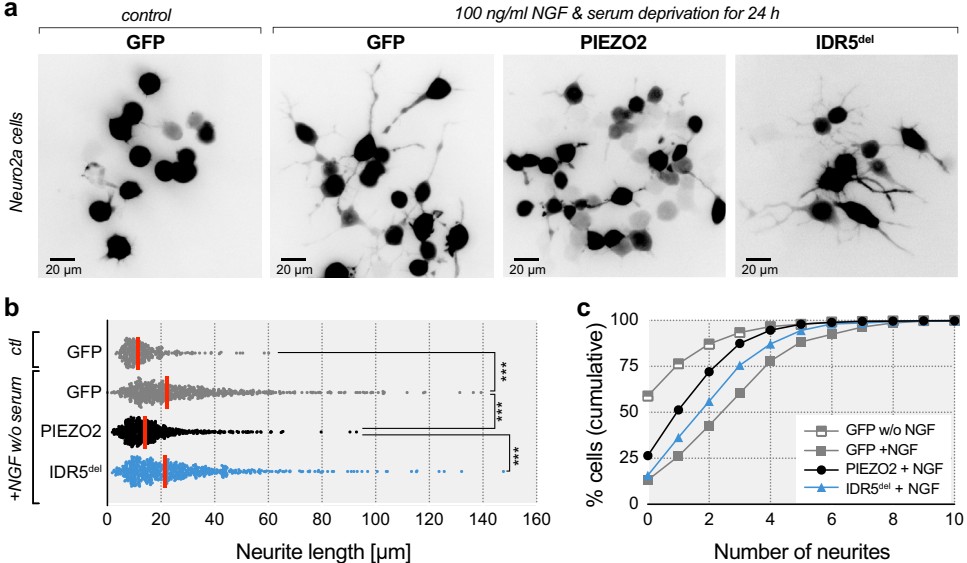

**Fig. 6 PIEZO2 but not IDR5$^{del}$ inhibits neurite outgrowth from N2a cells. a** Representative inverted fluorescent images of N2A-P1KO transfected with GFP vector in control medium (left), GFP, PIEZO2 (middle) or IDR5$^{del}$ (right) in medium without serum and with nerve growth factor (NGF). **b** Representation of the longest neurite measured (in μm) per cell for GFP in control medium and GFP (gray), PIEZO2 (black) and IDR5$^{del}$ (blue) in medium without serum and with NGF. Comparison with Kruskal–Wallis test, $p < 0.0001$, and Dunn's post-test: GFP control vs. GFP + NGF w/o serum ($P < 1e{-}15$), GFP + NGF w/o serum vs. PIEZO2 ($P < 1e{-}15$), PIEZO2 vs. IDR5$^{del}$ ($P < 1e{-}15$). Symbols are individual cells, with red line being the median. N-number of cells per group is: GFP control ($N = 343$), GFP ($N = 743$), PIEZO2 ($N = 733$), IDR5$^{del}$ ($N = 685$). **c** Cumulative distribution of the number of neurites per cell across GFP control, GFP, PIEZO2 and IDR5$^{del}$ group. Total number of cells per group is: GFP control ($N = 836$), GFP ($N = 854$), PIEZO2 ($N = 998$), IDR5$^{del}$ ($N = 810$). Source data are provided as a Source Data file.

with microtubule polymerization, on PIEZO2 and IDR5$^{del}$-mediated currents. Interestingly, nocodazole neither affected poking-evoked currents nor pressure-induced PIEZO2 and IDR5$^{del}$ currents, respectively (Fig. 5d and Supplementary Fig. 7a, b). We did, however, observe a small, yet statistically insignificant, reduction in the total charge transfer of pressure-induced currents (Fig. 5e–h) after nocodazole treatment.

The observation that Cyto-D treatment reduces poking-evoked PIEZO2 but not IDR5$^{del}$ currents, suggested that IDR5 is required for the detection of cytoskeleton-transmitted forces. Our experiments did, however, not clarify whether these forces directly activate PIEZO2 (force-from-filament) or if they solely facilitate membrane stretch-induced activation (force-from-lipids-via-filament) – note, membrane stretch inevitably occurs when poking a cell (Fig. 5i).

**PIEZO2 inhibits neurite outgrowth via an IDR5-dependent mechanism.** The identification of a mutant that inhibited cytoskeleton-dependent but not cytoskeleton-independent activation of PIEZO2 – i.e., IDR5$^{del}$ – enabled us to examine how naturally occurring stimuli in intact cells activate PIEZO2. An important function of PIEZO channels is the detection of cell-generated forces that occur during cell migration and neurite outgrowth[12,14–17]. We thus asked if PIEZO2 modulates neurite outgrowth from N2a cells and if so, how the cell-generated forces activate the channel.

Consistent with previous reports[55,56], we found that 24 h treatment with nerve growth factor (NGF, 100 ng/ml) and serum deprivation significantly increased the number of neurite-bearing cells from 41% (343/836 cells) to almost 87% (741/854 cells) amongst cells transfected with green fluorescent protein (GFP, Fig. 6a–c). Notably, NGF-treatment more than doubled the mean neurite length from $13.6 \pm 0.5$ μm (mean ± SEM, $N = 343$) to $28.7 \pm 0.8$ μm (mean ± SEM, $N = 743$) and significantly increased the average number of neurites per cell (Fig. 6a–c). Interestingly,

in cells expressing full-length PIEZO2, NGF-treatment and serum-deprivation only had a negligible effect on neurite outgrowth. Thus, although the number of neurite-bearing cells increased to 73% (733/998 cells) with NGF-treatment, the neurites of PIEZO2-expressing cells were only slightly longer ($17.4 \pm 0.5$ μm, $N = 733$) than those of untreated GFP-expressing cells and significantly shorter than those of NGF-treated and serum deprived GFP-expressing cells. Moreover, the number of neurites per cell was also smaller amongst PIEZO2-expressing cells as compared to GFP-expressing cells (Fig. 6c). Strikingly, IDR5$^{del}$ did not inhibit neurite outgrowth. Thus, NGF-treated and serum-deprived IDR5$^{del}$-expressing cells were almost indistinguishable from GFP-expressing cells with respect to neurite length ($26.3 \pm 0.8$ μm, $N = 685$, Fig. 6a, b), number of neurites per cell (Fig. 6a, c) and proportion of neurite-bearing cells (84%, 683/810 cells), suggesting that PIEZO2 requires IDR5 for the detection of cell-generated forces or other cues that might occur during neurite outgrowth.

## Discussion

The overarching goal of this study was to examine the role of the intrinsically disordered intracellular domains of PIEZO2, which account for one fourth of the channel and the function of which has hitherto not been investigated. While our study eventually focused on the role of IDR5, the initial characterization demonstrated that IDR2, -3 and -4 are also involved in fine-tuning the function of PIEZO2.

An important finding of this study was that IDR2 controls the velocity sensitivity of PIEZO2. Velocity sensitivity is a key feature of rapidly-adapting low-threshold mechanoreceptor afferents (RA-LTMRs), which encode the velocity rather than the amplitude of tactile stimuli by their action potential firing frequency[57–60] and utilize PIEZO2 as the primary mechanotransducer[6]. Rugiero and colleagues, who had described the velocity sensitivity of the rapidly-adapting mechanotransduction current in dorsal root ganglion

neurons, which was later shown to be mediated by PIEZO2, proposed that the velocity-dependence of LTMRs might result from the rapid inactivation of the channel[43]. They hypothesized that when cells or afferents are stimulated with slow stimuli, many channels close before the maximum stimulation magnitude is reached, so that not all channels contribute to the peak current amplitudes at the end of the stimulus ramp. Our data contradicts this hypothesis, firstly, because PIEZO1 inactivates much slower than PIEZO2 but produces even smaller currents at slow velocities (Fig. 3a, b) and secondly, because the deletion of IDR2 from PIEZO2 did not affect the inactivation kinetics (Fig. 1e), while it substantially changed velocity sensitivity (Fig. 3a, b). While our data highlights an important role for IDR2 in fine-tuning the velocity sensitivity of PIEZO2, it does not allow any conclusions about the mechanistic basis of this important property. Another interesting finding of our study was that deletion of IDR3 significantly increases single-channel conductance, probably by dislocating the lateral plug from the lateral ion-conducting portals (Fig. 3c and d), which strongly supports previous studies from others and ourselves that had proposed that the lateral plug controls ion permeation of PIEZO channels[26,27]. Moreover, these findings suggest that in addition to the alternative splicing of exon 33, which encodes the lateral plug, modifications of IDR3 might also have an impact on single-channel conductance. We also show that deletion of IDR4 significantly increased stretch sensitivity of PIEZO2 (Fig. 2c). Since we did not follow up on this effect, however, we do not consider it worth further discussion. Considering that IDR4 is partly encoded by exon 18 and 19, which are missing in the PIEZO2 splice variants that are expressed in the lung and the bladder[39], it is nevertheless tempting to speculate that IDR4 might contribute to the tissue-specific functions of PIEZO2.

Regarding the role of IDR5, our data suggests that it is required for force transmission from the cytoskeleton to PIEZO2. So how do these findings fit to the current concepts about PIEZO gating? The force-from-lipids gating model proposes that membrane stretch causes changes in the transbilayer pressure profile, which supposedly leads to conformational changes and thus activation of the channel[2,3,34,35]. Previous studies have demonstrated that PIEZO1 is indeed inherently sensitive to force-from-lipids, as it responds to stretch in lipid droplets[61]. Others, however, also showed that complete or partial detachment of the membrane from the cytoskeleton by blebbing[50] or genetic deletion of Filamin-A[62], respectively, as well as pharmacological disruption of the cytoskeleton[52,63] render PIEZO1 more sensitive to membrane stretch, suggesting that PIEZO1 is predominantly activated by force-from-lipids in cell-attached recordings and that the cytoskeleton solely has a negative modulatory effect on stretch sensitivity. Disruption of the actin cytoskeleton also increased the stretch sensitivity of PIEZO2 (Fig. 5e), indicating that the cytoskeleton has a similarly inhibitory influence on PIEZO2 activation, which together with the fact that PIEZO2 can be activated by both positive[64] and negative pressure suggests that PIEZO2[22] (Fig. 2) is predominantly activated by force-from-lipids in cell-attached pressure clamp recordings. IDR5, however, does not seem to be involved in the modulation of PIEZO2 stretch sensitivity by the cytoskeleton, as its deletion did not alter the stretch sensitivity of PIEZO2 in cell-attached recordings (Fig. 2).

In contrast to the force-from-lipids model, the force-from-filament model proposes that mechanical forces are transmitted to the channel by the cytoskeleton. Indeed, several studies as well as our own work have shown that an intact cytoskeleton is essential for the activation of PIEZO1[32,50,52] and PIEZO2[33,51] (Fig. 5a, b) in whole-cell poking experiments. Whether cytoskeleton-transmitted forces are the sole forces that activate PIEZO2 in whole-cell recordings or whether they act synergistically with membrane stretch, which inevitably occurs when poking a cell, is, however, unclear. A possible synergistic mechanism that has been put forward by several researchers is that cytoskeletal strain generates local membrane tension around the channels, such that the channel is eventually activated by membrane stretch – i.e., force-from-lipids-via-filament gating[34,35,65] (Fig. 5i). This gating model implicates that any PIEZO2 mutant with normal force-from-lipids sensitivity, would (i) also respond normal to poking in cells with an intact cytoskeleton and (ii) that disruption of the cytoskeleton would reduce the poking-evoked currents of such mutants. IDR5$^{del}$, which seems to have normal force-from-lipids sensitivity as it responds normal to stretch in cell-attached recordings, however, meets neither of these conditions, as it (i) produces only tiny poking-evoked currents (Fig. 1) and is (ii) completely insensitive to perturbations of the cytoskeleton (Fig. 5a, b). Hence, our data argues against the force-from-lipids-via-filament gating model. Another possible synergistic mechanism is that forces transmitted to PIEZO2 via the cytoskeleton and IDR5 normally facilitate membrane stretch-induced activation of PIEZO2. The fact that deletion of IDR5 did not alter stretch-sensitivity, however, also argues against this hypothesis.

Hence, we propose that PIEZO2 is activated by cytoskeleton-transmitted forces and that IDR5 – specifically the negatively charged AA stretch E625–E630 – is required for the transmission of force from the cytoskeleton to the channel. The observation that disruption of the cytoskeleton and deletion of IDR5 markedly but not completely abolished poking-evoked PIEZO2 currents together with the observation that IDR5$^{del}$ is resistant to Cyto-D treatment, however, also suggests that a small proportion of the channels that mediate poking-evoked whole-cell currents is activated by a cytoskeleton-independent mechanism – possibly poking-evoked membrane stretch. At present, we can only speculate about why some channels are activated by cytoskeleton-transmitted forces and others, within the same cell, by membrane-stretch in whole-cell recordings. Considering that membrane tension is only propagated a short distance across the cell surface[66], it is tempting to speculate that sufficient membrane stretch is only exerted on channels that are located close to the site of mechanical stimulation, whereas channels that are located further away are predominantly exposed to forces transmitted by the cytoskeleton. An alternative explanation is that channels that are localized in clusters that exhibit confined movement or sub-diffusion are more likely to be tethered to the cytoskeleton and may thus be activated by cytoskeleton-transmitted forces, whereas channels from clusters that show normal lateral diffusion might be activated by changes in membrane tension. At first glance, our observation that IDR5$^{del}$ is less sensitive to cytoskeleton-transmitted forces than full-length PIEZO2 while both channels show similar lateral diffusion characteristics, argues against this hypothesis. It is, however, possible that IDR5 is merely required for the PIEZO2 activation by cytoskeleton-transmitted forces and that the actual physical tethering is mediated by other domains or by a separate protein. Moreover, it is possible that individual channels (i.e., channels that are not part of any cluster), which are not resolved by TIRF imaging and which have recently been show to function as independent mechanotransducers[67], are preferentially activated by one of the two force transmission pathways and that their mobility or tethering is affected by the deletion of IDR5. Demonstrating the latter hypothesis would, however, require time-lapse live-cell imaging at super-resolution, which is far from being a well-established technique.

The recurring observation that certain domain deletions (IDR2, IDR5 and IDR4) had differential and very specific effects on poking-evoked and pressure-evoked PIEZO2 currents, demonstrate that the full extent of the functional deficit of a channel variant and thus mechanistic insights into channel

function can only be revealed when the full repertoire of electrophysiological techniques is employed. Hence, our results have far-reaching implications for the design and interpretation of future PIEZO channel studies.

While the question how PIEZO2 is gated in different patch-clamp assays is very important, the biologically more relevant question is how PIEZO2 is activated by naturally occurring stimuli. In this respect, the observation that IDR5[del] responds normal to membrane stretch (Fig. 2) but fails to inhibit neurite outgrowth (Fig. 6) is very interesting, because it indicates that forces acting on or in the cells during neurite outgrowth probably activate PIEZO2 via a cytoskeleton-dependent mechanism and, moreover, highlights the importance of examining PIEZO channel function in fully intact cells and independent of electrophysiological assays before drawing conclusions about the physiological relevance of certain channel mutations or splice variants. Hence, PIEZO2 adds to the list of mechanically-gated ion channels that require interactions with the cytoskeleton for normal function, such as NOMPC, TMC1, ENaC, MEC4 and MEC10[37,38,53,54]. An important question that remains open and calls for further investigation is whether IDR5 directly links PIEZO2 to the cytoskeleton or if a separate tether protein mediates this interaction.

## Methods

**Cell culture and transfection.** Neuro2A PIEZO1-Knockout cells (N2A-P1KO, gift from G.R Lewin[25]) were grown at 37 °C with 5% CO$_2$ in Dulbecco's Modified Eagle Medium (DMEM) and optimal Minimal Essential Medium (opti-MEM) (1:1 mixture) with 10% Fetal Bovine Serum (FBS), 2mM L-glutamine and 1% penicilline/streptomycine (all from Thermo Fisher). Cells were seeded on poly-L-lysine (PLL, Sigma) coated glass coverslips (for patch-clamp and immunocytochemistry), PLL and laminin-coated coverslips (Neuvitro, for neurite imaging), 35 mm 6 well plates (biotinylation assay) or PLL-coated 35 mm glass-bottom dishes (TIRF microscopy live-imaging). N2A cells were transfected one day (neurite imaging, biotinylation) or two days (patch-clamp, immunohistochemistry, live-imaging) after plating using polyethylenimine (PEI, Linear PEI 25 K, Polysciences). For one coverslip, 7 µl of a 360 µg/ml PEI solution is mixed with 9 µl PBS. Plasmid DNA is diluted in 20 µl PBS (0.6 µg/coverslip) and then added and mixed to the 16 µl PEI-PBS solution. After at least 5 min of incubation, 36 µl are added in one well and mixed by gentle swirling. For a 35 mm well or dish, 1.5 µg DNA is used and PBS/PEI volumes are adjusted accordingly. 24 h later, the medium is replaced by fresh one. Cells are then used within 24 h (neurite imaging, biotinylation, patch-clamp, live-imaging) to 48 h (patch-clamp, immunohistochemistry, live-imaging).

**Constructs and generation of PIEZO mutants and chimera.** A PIEZO2-HA-IRES-GFP plasmid was previously created[27] from mouse piezo2-pSPORT6 plasmid (gift from A. Patapoutian) by adding at the C-terminus an HA-IRES-GFP sequence with BamHI/NotI restriction sites and was used as the initial template to generate all the construct of the present study. For IDR deletion mutants, two AfeI restriction sites were sequentially added by two rounds of PCR-amplification on each side of individual IDR (see Fig. 1b and Supplementary Fig. S2 for amino acid position) using KAPA HiFi polymerase (Roche). A similar approach was used to generate the PIEZO1 IDR chimera, where two restrictions sites flanking the corresponding PIEZO1 IDR were first introduced. The PIEZO2-HA-IRES-GFP-IDR3 + LateralPlugdel was generated with homologous recombination from IDR3del (NEBuilder HiFi, New England Biolabs). Point mutations (polyA) were introduced by PCR. To generate PIEZO2mScarlet constructs, mScarlet was first codon-optimized to remove a NotI restriction site. Then the HA-IRES-GFP sequence from PIEZO2 and IDRdel was excised and replaced with PCR-amplified mScarlet and ligated using BamHI/NotI restriction sites. All the corresponding primers are listed in Supplementary Table 1. PCR reactions were DpnI digested (New England Biolabs, 37 °C, 1 h) and column purified (Macherey-Nagel) before being transform in electrocompetent Stbl4 bacteria (Invitrogen) and grown at 30 °C for 48 h. After the two restriction sites were incorporated, plasmids were digested with AfeI (PIEZO2) or AgeI (PIEZO1) (both from New England Biolabs) and gel purified to remove the cut IDR fragment. For PIEZO1 chimera, the corresponding PIEZO2 IDR fragment was PCR amplified and then ligated. The IDR2 fragment excised in PIEZO1 was D1576-E1658, replaced by L1731-D1961. Ligations were performed overnight at 16 °C (re-ligation for PIEZO2-IDRdel) or for 2 days at 4 °C for the others (ligase from Promega) and then transformed in Stbl4 bacteria. Selected clones were entirely sequenced to ensure that no other mutation was present.

**Neurite outgrowth assay.** N2A-P1KO cells were prepared as described above. Cells were plated at a density of 5000–10,000 cells per well containing one 12 mm PLL + laminin-coated coverslip (Neuvitro) and processed for transfection one day later. 24 h after transfection, cells were incubated with N2A medium without FBS and with Nerve Growth Factor (NGF, 100 ng/ml, Sigma) to induce neurite outgrowth. Negative control coverslips were incubated instead with standard N2A medium. Neurites were imaged 30–40 h after using the GFP fluorescent signal from PIEZO transfected cells. An empty GFP vector was also used separately as a control. Fluorescent live images were acquired on an inverted microscope (IX70, Olympus) with a 20x oil-immersion objective and visualized with an Imago-QE-Sensicam camera (PCO). Approximately 20 images were acquired per coverslips. One to three coverslips per constructs were used. The results presented here come from three independent experiments and transfection. Image analysis was manually performed with Fiji (Version 2.3.0/1.53f), using the Region Of Interest (ROI) function to count and measure neurites. GFP-positive cell body perimeter was first marked, followed by the neurite and the side-branches.

**Whole-cell patch-clamp recordings.** Mechanically activated currents were recorded at room temperature using EPC10 amplifier (HEKA) with Patchmaster (Version 2×91) and Fitmaster (Version 2×91) software (HEKA). The borosilicate patch pipettes (2–5 MΩ) were pulled with a Flaming-Brown puller (Sutter Instruments) and contained the following (in mM): 125 K-gluconate, 7 KCl, 1 MgCl2, 1 CaCl2, 4 EGTA, 10 HEPES, 2 GTP and 2 ATP (pH 7.3 with KOH). The control bath solution contained the following (in mM): 140 NaCl, 4 KCl, 1 MgCl$_2$, 2 CaCl2, 4 glucose and 10 HEPES (pH 7.4 with NaOH). Cells were held at a holding potential of −60 mV and stimulated with a series of 13 mechanical stimuli in 0.4 µm increments with a fire-polished glass pipette (tip diameter 2-3 µm) that was positioned opposite to the recording pipette, at an angle of 45° to the surface of the dish and moved with a velocity of 1 µm/ms by a piezo-driven micromanipulator (Nanomotor© MM3A, Kleindiek Nanotechnik). The evoked whole-cell currents were recorded with a sampling frequency of 200 kHz and filtered with 2.9 kHz low-pass filter. Pipette and membrane capacitance were compensated using the auto function of Patchmaster. Leak currents before mechanical stimulations were subtracted off-line from the current traces. Recordings with excessive leak currents, unstable access resistance and cells which giga-seals did not withstand at least 7 consecutive mechanical steps stimulation were excluded from analyses.

The mechanical thresholds of the PIEZO2-mediated currents were determined by measuring the latency between the onset of the mechanical stimulus and the onset of the mechanically activated current. Current onset was defined as the point in which the current significantly differed from the baseline ($< [I_{mean*baseline} - 6 \times SD_{baseline}]$). The membrane displacement at which the current was triggered was then calculated by multiplying the speed at which the mechanical probe moved (1 µm/ms) with the latency. The inactivation time constants ($\tau_{inact}$) were measured by fitting the mechanically activated currents with a single exponential function ($C1 + C2*exp(-(t - t0)/\tau_{inact})$, where C1 and C2 are constants, $t$ is time and $\tau_{inact}$ is the inactivation time constant. I/V curves and $E_{Rev}$ were determined by changing the holding potential in −30 mV steps (−60 to +60 mV) and by stimulating the cells with a fixed mechanical displacement that evoked a submaximal response.

The velocity-dependance of activation of PIEZO currents was tested by submaximal mechanical stimulations of the cell (3.6 µm displacement, every 10 s), first at 1 µm/ms until stable current responses were observed and then by progressively decreasing the velocity of stimulation (0.75, 0.5, 0.25, 0.125, 0.0625, 0.0312 and 0.0156 µm/ms). Cells with unstable responses (coefficient variation of the amplitude of the last 3 responses superior to 0.2) at the standard 1 µm/ms velocity were excluded.

To investigate the contribution of cytoskeleton component in PIEZO currents, N2A cells were incubated prior recordings with Cytochalasin-D (1 µM) for 15 min, Nocodazole (1 µM) for 30 min (both from Sigma) or Jasplakinolide for 1 h (200 nM, Tocris). All were dissolved in DMSO (final concentration ≤ 0.1%). The drugs were also kept in the standard electrophysiological bath solution during the experiments.

**Single-channel recordings.** Single-channel stretch-activated currents were recorded in the cell-attached configuration at room temperature using EPC10 amplifier (HEKA) with Patchmaster (v2x91) and Fitmaster (v2x91) software (HEKA). The borosilicate patch pipettes were coated with Sylgard (WPI) and fire polished (final resistance of 4–8 MΩ). The pipette solution contained the following (in mM): 130 NaCl, 5 KCl, 1 MgCl$_2$, 1 CaCl$_2$, 10 HEPES, 10 TEA-Cl (pH 7.3 with NaOH). The bath solution contained (in mM): 140 KCl, 1 MgCl$_2$, 2 CaCl$_2$, 10 Glucose, 10 HEPES (pH 7.4 with KOH). Pressure stimuli were applied with a 2 ml syringe operated by a motorized device and measured with a custom-made pressure sensor[27] or with the High-Speed Pressure Clamp (HSPC, ALA scientific). The evoked current were recorded with a sampling frequency of 50 kHz and filter with a 2.9 kHz low-pass filter. Pressure-response curves were evoked by a stepwise increase of negative pressure (3 s duration) with the cell being clamped at a holding potential of −100 mV. In response to repetitive and sustain pressure pulses, especially over −20 mmHg, PIEZO2 has the tendency to produce non-inactivating responses, making the determination of a potential "peak current" value difficult or impossible in most of the cases. Therefore, to accurately quantify stretch-activated PIEZO2 currents, we calculated the total charge transferred during the pressure

stimulus (in pico Coulomb, pC) through the determination of the area under the curve over the 3 s stimulus. Single-channel amplitudes at a given holding potential ($-120$ mV to $-40$ mV, 20 mV steps) were determined as the difference between the peaks of the Gaussian fits of the trace histogram over a 500 ms segment. Pressure used to evoke channels response in those experiment was selected to produce distinguishable single openings and adjusted from cell to another. Unitary conductance was determined from the linear regression fits of the I/V plot of individual cells. Recordings with excessive leak currents (>4 pA) or unstable baseline were excluded from analyses. Recordings that displayed non-inactivating responses or unstable openings were not used for I/V analyses. The effects of cytoskeletal modifying drugs were tested in the same conditions as for the whole-cell experiments.

**Immunocytochemistry**. N2A cells were co-transfected with PIEZO2-HA-IRES-GFP or IDRdel constructs and with a plasmid encoding the red fluorescent protein mScarlet fused to a farnesylation signal sequence in its C-terminus to target it to the plasma membrane. Three days after transfection, cells were washed once with PBS and fixed with 4% PFA for 10 min at room temperature, washed 3 times for 5 min with PBS and permeabilised for 1 h at room temperature (permeabilization buffer: 2,5% donkey serum (Sigma), 1% BSA, 0.1% Triton X-100, 0.05% Tween-20, in PBS). Samples were then incubated overnight at 4 °C with a 1:500 dilution of rabbit anti-HA antibody (Thermo Fisher Scientific) in PBS 1% BSA. After 3 washes of 5 min, cells were incubated for 1 h with a 1:1000 dilution of AlexaFluor-647 donkey anti-rabbit (Life technologies, diluted in PBS 1% BSA) and washed 3 more times. Coverslips were mounted on slides with FluoProbe mounting media that contain DAPI (Interchim). Confocal images were acquired with a SP8 confocal microscope (Leica) and a 63x oil-immersion objective. Images were analyzed off-line with Fiji (v2.3.0/1.53f).

**TIRF microscopy and live-imaging**. Live imaging was performed on a Nikon Ti2 microscope and Nikon H-TIRF module using NIS Elements software (v6.14, Nikon). The objective was an oil immersion Nikon Apo TIRF 100x (NA 1.45). A 1.5x magnification lens was added giving a final pixel size of 0.11 µm. The TIRF angle was adjusted manually for every cell if necessary. The camera used was an Andor iXon Ultra DU-897U single photon detection EMCCD, having a resolution of 512 × 512 pixels. Cells were illuminated with a 561 nm excitation laser and were imaged for 30 s with a frame rate of approximately 10 Hz (100 ms exposure time). An on-stage incubation chamber (TokaiHit) was used to control temperature (37 °C), $CO_2$-concentration (5%) and humidity. N2A cells transfected with PIEZOmScarlet constructs were prepared as described above and imaged in phenol-red free medium.

Time-lapse recordings were preprocessed in Fiji (v2.3.0/1.53f) before track analysis, with a background subtraction and a bleach correction (Histrogram matching). PIEZOmScarlet track analysis was then performed with TrackMate v6.0.1[48] and the following parameters: DoG detector, blob diameter of 0.7 µm, spots quality filter value of 14 to 15, simple LAP tracker with a linking distance of 0.5 µm, a gap closing distance of 0.7 µm and a maximal gap closing frame number of 2. To ensure accurate calculation of MSD, only tracks that have a duration of at least 40 frames were considered. Further track classification was done with TraJClassifier[49]. MSD calculation was then performed in Igorpro 8 (WaveMetrics). To analyze the diameter of PIEZO cluster and their density per cell, the total number of spots detected by TrackMate on the first frame was used and the cell area was manually determined in ImageJ. Individual spot diameter calculation was performed in IgorPro by using the XY coordinates of every spot and by fitting them with a 2D Gauss function. The corresponding spot diameter is equaled to 2× the standard deviation.

**Membrane biotinylation assay**. N2A cells were cultured and transfected as describe above. Two days after transfection, cells were washed twice with ice-cold PBS and treated for 30 min with 0.6 mg/ml of biotin in PBS (EZLink Sulfo-NHS-LC-Biotin, Thermo Scientific). Next, an equivalent volume of glycin solution (100 mM in PBS) was added to quench the remaining biotin. After 30 min, cells were washed twice with cold PBS and resuspended in 1 mL of PBS. To harmonize the number of cells between samples, turbidity at 600 nm was measured. Cells were then centrifuged for 2 min at $1700 \times g$ and lysed with RIPA buffer (50 mM Hepes, pH 7.4, 140 mM NaCl, 10% glycerol, 1% (v/v) Triton X-100, 1 mM EDTA, 2 mM EGTA, 0.5% deoxycholate, 10 mM PMSF and protease inhibitor mixture (Roche)) under gentle agitation for 1 h at room temperature. Volume of RIPA buffer was by default 200 µl for the sample having the highest optical density. The volume for the other samples was adjusted accordingly to obtain a similar cell concentration. Extracts were centrifuged at $12,900 \times g$ for 15 min and the supernatant was kept. 10 µl of supernatant was taken and diluted with 10 µl of 4× Laemmli buffer (total extract fraction). Equilibrated Streptavidin resin (High Capacity Streptavidin Agarose, Thermo Fisher) was added to equal amounts of the remaining super-natants and incubated under gentle agitation overnight at 4 °C. After washing, biotinylated proteins were eluted by boiling the resin in 30 µl of 2× Laemmli buffer for 5 min (biotinylated/membrane fraction), and subsequently analyzed by SDS-PAGE and western blot.

**Western blotting**. Biotinylated eluates and the corresponding total extract fraction were electrophoresed in 10% polyacrylamide gels. Next, they were blotted onto a nitrocellulose membrane (0.4 µm, GE Health care) using a transfer buffer consisting of 30 mM Tris base, 190 mM glycine, and 20% methanol. After blocking at room temperature in TBS-T (20 mM Trizma base, 500 mM NaCl, 0.1% Tween-20) with 2% skimmed milk powder for at least 30 min, membranes were incubated overnight with rabbit anti-HA (Sigma, dilution 1:1000 in TBS-T 5% skimmed milk) or rabbit anti-Beta tubulin (Sigma, used as a loading control, dilution 1:5000 in TBS-T 5% skimmed milk) at 4 °C. After washing with TBS-T, membranes were incubated with the secondary antibody for 1 h at room temperature (Sigma, 1:10000, anti-rabbit HRP conjugated, diluted in TBS-T 5% skimmed milk). Finally, the immunoreactive bands were revealed using ECL Prime (GE Healthcare), visualized with the iBright1500 system and further analyzed with the iBright Analysis software (v3.0.1; Invitrogen). The results presented here come from four independent experiments.

**Structural modeling**. To enable a better visualization of the location of the IDRs, the structures of which were not resolved in the recently published cryo-EM structure, structural models of the IDRs were generated with the SWISS-MODEL protein structure homology-modeling server using the published PIEZO2 cryo-EM structure (6KG7[22]) as template. The modeled IDR structures were subsequently combined with original PIEZO2 cryo-EM structure (6KG7) to create a structural model of full-length PIEZO2. Note, the QMEAN-values of the intracellular domains, which indicate the model quality, were extremely low and thus the structural models of the IDRs solely serves as an orientation guide that should give an idea about the approximate size and localization of the intracellular domains. All molecular images of PIEZO2 were generated with PyMOL 2.4.0 (Schrödinger, LLC). The mouse PIEZO2 amino acid sequence composition and properties were analyzed with Jalview 2.11.0. Disorder prediction was performed on mouse PIEZO2 sequence with PONDR (VSL2), IUPred and ESpritz[40–42].

**Data analysis**. All electrophysiological data were analyzed using FitMaster (version 2 × 91, HEKA) and IgorPro 8 (Wavemetrics). All imaging data were analyzed using the Fiji image analysis package.

**Statistics**. Unless otherwise stated, all data are expressed as means ± s.e.m. All statistical analyses were performed with Excel and Prism 8.0 (Graphpad). Data distribution was systematically evaluated using D'Agostino–Pearson test and parametric or non-parametric tests were chosen accordingly. The statistical tests that were used, the exact P-values and information about the number of independent biological replicates are provided in the display items or the corresponding figure legends. Symbols on graphs (* or #) indicate standard P-value range: *$P < 0.05$; **$P < 0.01$; ***$P < 0.001$ and ns (not significant) $P > 0.05$. Additional information about the statistical tests can be found in a separate statistics information file, which is provided with this paper.

**Reporting summary**. Further information on research design is available in the Nature Research Reporting Summary linked to this article.

## Data availability

All data supporting the findings of this study are available within the article and its supplementary information files. Additional information, relevant data (electrophysiological and imaging raw data) and materials (plasmids encoding the PIEZO channel mutants generated in this study) are available from the corresponding author upon reasonable request. Source data are provided with this paper.

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

## Acknowledgements

This study was supported by the DFG grants LE3210/3-1 and SFB1158/1 to S.G.L. We thank Ms. Anke Niemann for technical assistance. We also thank Dr. Ulrike Engel for help with setting up the TRIF experiments at the Nikon Imaging Facility at Heidelberg University.

## Author contributions

C.V. cloned PIEZO2 mutants, performed and analyzed patch-clamp recordings, TIRF experiments, biotinylation assays and neurite outgrowth assay and wrote the paper. I.S. cloned PIEZO2 mutants, performed and analyzed patch-clamp recordings. F.J.T. cloned

PIEZO2 mutants. J.M.J. and T.A.N. performed and analyzed neurite outgrowth assays. N.W. performed immunocytochemistry. S.G.L. conceptualized the study, acquired funding, analyzed the data, supervised the project and wrote the manuscript.

## Funding

## Competing interests
The authors declare no competing interests.
