## [Peer Review File · Nature Communications]

Intrinsically disordered intracellular domains control key features of the mechanically-gated ion channel PIEZO2REVIEWER COMMENTS

Reviewer #1 (Remarks to the Author):

A central question to force-gated ion channel function is uncovering the underlying mechanisms of force transmission. In this study Verkest and colleagues address the potential for cytoskeletal force transmission in Piezo2 by systematically determining the role of several large intracellular linker domains that have thus far remained unresolved in available cryo-em structures using two independent assays: pressure-clamp and cell-poke.

The authors find a single linker (IDR5) that drastically reduces poke responses of Piezo2, while minimally affecting on responses by pressure-clamp. Moreover, they show that the sensitivity Piezo2 poke responses to the actin destabilizing agent Cytochalasin D depend on this domain. From this the authors conclude that the IDR5 domain directly interacts with the actin cytoskeleton to transmit mechanical forces and drive long-range channel gating.

While this finding is truly exciting, it is not sufficient to support this strong mechanistic conclusion. A number of critical experiments and conceptual developments are absolutely required to support the conclusion of a tether mechanism:

1) The authors interpretation is entirely depending on the idea that force transmission in pressure-clamp is primarily mediated by force-from-lipids, whereas in poke there is a large recruitment of long-range channels via force-from-filament. However, this framework is far from established and complicated by a number of factors:

a) Contrary to what the authors claim, it has been previously demonstrated (Cox 2016) and is also evident from the manuscript's own data (Figure 5e-g) that the cell-attached configuration, and even inside-out patches, are not devoid of cytoskeleton or cytoskeletal influence.

b) The stretch sensitivity experiments are difficult to evaluate due to the inconsistent current responses and the fact that the stretch response of Piezo2 does not saturate prior to membrane lysis.

c) The authors state the differences seen in whole-cell poke are due to the loss of long-range activation via force-from-filament, however no such long-range activation was ever shown. Even taken at face value, as demonstrated in Shi 2018, tension should diffuse over a wider area upon cytoskeletal disruption in addition to the potential of any lost long-range contacts complicating this interpretation.

In order to address all of these points the authors have to demonstrate, using an imaging or equivalent approach, that long-range activation of Piezo channels, and subsequent loss upon CytoD treatment IDR5 deletion, is actually occurring.

2) There are well-established approaches to determining a direct cytoskeletal interaction that should be performed in order to prove a tether mechanism: If IDR5 does indeed interact with a tethering protein the authors should be able to:

a) Identify the tethering protein.

b) Show in pull-down experiments that the tethering protein and Piezo2 interact.

c) Perform competition assays to validate the functional interaction.

Minor concerns:

1) The analyses in figure 3 should be limited to patches that show distinct single-channel openings. Geng et al. observed multiple subconductance states upon substitution of the residue corresponding to Y1568 immediately preceding IDR3. It is possible that the removal of this large proximal domain would have similar effects. Performing the analyses on patches showing overlapping channel openings prevents the identification of these states and they may spuriously combine to suggest full single-channel openings.

2) It is inappropriate to refer to the intracellular domains as “disordered regions”. The term disordered makes specific inferences about the structure of these regions. Their absence in available structures implies that they are likely to be flexible, but perhaps not necessarily disordered. A nice example of why this nomenclature is not appropriate is seen in TRPA1 channels, where the ankyrin repeat domain is difficult to resolve, but extremely likely to be highly structured.

3) Figure 1a includes structural models of the IDR regions, however it is unclear how these models were generated. This should be included in the methods and the fact that these models were generated and are not a part of the original structure should be made clearer when describing the figure.

4) (pg 12 line 16) the authors write: “This is very interesting, because a part of IDR4 is encoded by exon18, which is missing in Piezo2 splice variants that are expressed in the lung and bladder where Piezo2 is mostly exposed to stretch rather than focal indentation of the cell membrane. Hence, a role of IDR4 in fine-tuning Piezo2 stretch-sensitivity would make perfect sense.” However, this line of reasoning is not well supported. Even in the skin the majority of stimuli will deform the tissue on a scale much larger than the size of a cell resulting in stretch as opposed to a focal indentation. The authors should tone down or remove this statement.

Reviewer #2 (Remarks to the Author):

An intrinsically disordered intracellular domain of PIEZO2 is required for force-from-filament activation of the channel

Verkest...Lechner Nature Communications

In the present manuscript, the authors aim to tackle PIEZO structure-function. This is an interesting area of study, particularly because the authors focus on structure-function of PIEZO2, which is less understood than PIEZO1. Given that splice variants of PIEZO2 are differentially expressed in known mechanosensitive tissues (bladder, lung, sensory neurons), and that PIEZO2 missense mutations can have a broad range of consequences for humans, it is crucial to uncover how structural domains of PIEZO2 govern mechanical gating in response to different kinds of forces. The electrophysiology data provide a detailed study illuminating the role of intrinsically disordered regions of mouse PIEZO2. This dataset will substantially advance our knowledge of PIEZO2 structure-function. Additionally, the data bring up the important point that poke and stretch, the two assays mainly used to study mechanically activated ion channels, have differing effects on PIEZO2 deletion mutants lacking specific IDRs.

However, the authors go too far in their conclusions. A glaring inconsistency in the paper is the bold claim in the title and abstract that PIEZO2 is directly activated by cytoskeletal forces, with comparatively little data to back up such a strong claim. While their hypothesis is an interesting one, the IDR deletion studies do not adequately address it. To convincingly show force-from-filament, the authors would need to either specifically identify the tether molecule and salient PIEZO2 residues or demonstrate the interaction physically, as was nicely shown for NOMPC and its ankyrin repeats (Zhang et al 2015, Cell). Recommending such experiments would far exceed what would be permissible in a revision. We also note the recent study from Romero et al. on margaric acid dependent modulation of PIEZO2 (Nature Communications, 2020). They show that cytoskeletal disruption (with Latrunculin A) modulates PIEZO2 ion channel activity. However, they temper their conclusions by stating that this is evidence for “cytoskeletal regulation”. While the authors of this manuscript examine the IDRs as opposed to the beam, the level of evidence for cytoskeletal interaction of PIEZO2 is comparable. The title, abstract, and conclusions should be rephrased to support the data contained within. We have not combed through line-by-line as these changes will require a substantial rewrite of the manuscript text. These changes are essential for revision of the manuscript.

What I believe the authors do succeed in demonstrating is that the intrinsically disordered domains IDR3, 4, and 5 have different effects on poke versus stretch evoked currents. The findings suggest that there may be distinct functional domains of PIEZO2 that are attuned to specific mechanical stimuli and/or that there may be interacting proteins that facilitate these distinct modes of activation, cytoskeletal or otherwise. Any speculation as to the underlying mechanism should be presented as such, as the evidence is insufficient to demonstrate one way or the other. My issue with the authors’

conclusions is not merely a question of semantics: a crucial flaw in the logic of the authors is their underlying assumption that because there is a poke vs. stretch difference, that it must be due to cytoskeleton vs. bilayer activation. Figures 4 and 5 serve as “roundabout” experiments that do not have the potential to disprove the hypothesis. Rather, Figure 4 and 5 implicate a role for IDR5 allosteric modulation of PIEZO2 activity by PKA and affirm that IDR5 modulates the mechanosensitivity of PIEZO2. Figure 6 is confusing since it is unclear as to what are the precise mechanical underpinnings of neurite outgrowth in the NGF + serum deprivation assay in the N2A cells. However, that they see an effect of PIEZO2 transfection suggests that there is indeed a mechanical component to the process worthy of future investigation, and this should be discussed. What would be more convincing than Figures 4, 5, and 6 are PIEZO1 chimera experiments with the IDRs followed by poke and stretch (where the protein homology renders this feasible).

Irrespective of the flawed conclusions, this paper presents important structure-function data that will advance the mechanotransduction field, even if the wording is toned down substantially, as we recommend. Below I list specific experimental concerns and suggest some new experiments to partially address the conclusions as presented in the present manuscript.

Major Experimental concerns:

1. Mock transfected or untransfected controls must be shown for all electrophysiology experiments (poke, stretch, and single channel stretch), with comparable N. We realize that poke was shown in Moroni et al 2018, but the authors should include these for all the major electrophysiology panels in the first half of the paper (1d, 2b-c, 3b) to ensure rigor.
2. It was surprising to see that PIEZO2 could tolerate these IDR deletions and still retain function. In Figure 3G, surface biotinylation kits are notorious for labeling intracellular protein as well as extracellular. It would be better to use a more specific form of surface labeling, such as by engineering the tag onto an extracellular region and performing antibody staining in permeabilized vs. nonpermeabilized cells (Coste et al 2015 Nat Comm). Thus the claim that there is no trafficking defect is unfounded. The small changes in whole cell current for IDR5 could be explained equally well by impaired trafficking. Were this the case, it would entirely refute the mechanism proposed in the paper. This claim should be toned down or additional supporting data should be presented. While the IDR3 region clearly affects the pore, the mechanism is less clear for IDR5 and the paper hinges on there being no trafficking defect.
3. A concern of the cytochalasin D experiments is that by altering the cytoskeleton, one also indirectly alters the surface tension of the cell membrane, thus affecting poke and stretch sensitivity independently of proposed PIEZO2-cytoskeleton tethers. We like to assume in these experiments that we are only altering the cytoskeleton, but for this paper especially it is crucial to affirm whether that assumption holds. It would be prudent to quantify cell volumes and surface areas to ensure the authors

are accounting for any changes in cell surface tension that could indirectly link the cytochalasin D effects to IDR5 and PIEZO2 function. The authors should also discuss the role of cytoskeleton in cell attached vs. excised patch: is there no role of cytoskeleton in either? How much does cytoskeleton contribute to activation in cell attached stretch?

Suggestions for new experiments:

4. I would suggest PIEZO1-PIEZO2 chimera experiments (IDR swaps), similar to what was performed for the beam in Romero et al (see above). This would render the cytochalasin D experiments more convincing than a simple deletion.

5. And with regards to the deletion, typically we see a glycine linker added in these experiments rather than a deletion (see Wang et al 2019 Nature). It might be worthwhile since the constructs are already made, to perform a site directed insertion of GGG or GGGG into IDRs 3,4, and 5 and re-test the construct in poke and stretch.

Reviewer #3 (Remarks to the Author):

The manuscript by Verkest et al. examines the effect of intrinsically-disordered regions (IDRs) on Piezo2 channel function. Through a careful structure-function study, the authors identify interesting new roles for some IDR domains of the protein that shed light on gating mechanisms as well as modulation of channel function. The work is significant, timely and well executed, and will be of interest to researchers in the fields of mechanotransduction and ion channel biophysics. Several points need to be addressed before publication.

Major points:

The conclusion that Piezo2-mediated inhibition of neurite outgrowth relies on the detection of cell-generated traction forces by the channel is not substantiated by experimental findings. As such, more experiments are necessary to demonstrate that cellular traction forces activate Piezo2 through the force-from-filament mechanism mediated primarily by IDR5. Do Neuro2a cells with NGF and w/o serum show Piezo2-mediated Ca²⁺ signals on hard substrates, which are reduced on soft substrates? Are those Ca²⁺ signals abrogated by traction force inhibitors in full-length Piezo2 (e.g. Blebbistatin or ML-7) and reduced or absent in the IDR5del mutant channel? Is neurite outgrowth different on soft substrates or on hard substrates with traction force inhibitors?

The finding regarding the greater stretch sensitivity of IDR4del is very interesting. Can the authors comment on the possible underlying mechanisms?

Figure 3: The proposed mechanism for increase in single channel conductance with the IDR3del mutation is intriguing and would be strengthened by an experimental confirmation, for instance with a double deletion of the lateral plug.

Figure 4, Page 8: "our data suggest that the PKA dependent potentiation of PIEZO2 observed in our expression conditions does not require phosphorylation of the channel and is probably mediated by an indirect mechanism." While this is a possibility, it is also likely that the deletion of IDR5 affects phosphorylation of other sites. Thus, this claim should either be substantiated with direct biochemical evidence or significantly tempered.

Figure 5: Traces should also be shown for Piezo1 stretch activation of IDR5del with cytoD and Nocodazole (Fig. 5e). Also, does Nocodazole have an effect on IDR5del channels in poking mode (Fig. 5h)? It may also be helpful to examine the effect of an actin stabilizer, e.g. jasplakinolide, on channel activity.

Minor points:

Page 4, line29-30: Please summarize the evidence for intracellular tethering for NOMPC and TMC1

Page 10, line 25-26: Also cite Pathak et al. PNAS 2014, 111 (45) 16148-16153, which was the first report of Piezo channel activation by cellular traction forces.

In Figure 1, it would be useful to have a schematic of Piezo2 showing the IDR sites, similar to that in Fig. 4A.

Figure 2a, please also show a family of ionic current traces for different pressure levels.

Figure 3B: Single channel traces and amplitude histogram for IDR4del also should be shown. Additionally, the pressure used should be mentioned

For clarity, it would be helpful to make the two arrows of different colors in Fig. 3e, and to use different colors for the plug and the latch domains.

Reviewer #1 (Remarks to the Author):

A central question to force-gated ion channel function is uncovering the underlying mechanisms of force transmission. In this study Verkest and colleagues address the potential for cytoskeletal force transmission in Piezo2 by systematically determining the role of several large intracellular linker domains that have thus far remained unresolved in available cryo-em structures using two independent assays: pressure-clamp and cell-poke.

The authors find a single linker (IDR5) that drastically reduces poke responses of Piezo2, while minimally affecting on responses by pressure-clamp. Moreover, they show that the sensitivity Piezo2 poke responses to the actin destabilizing agent Cytochalasin D depend on this domain. From this the authors conclude that the IDR5 domain directly interacts with the actin cytoskeleton to transmit mechanical forces and drive long-range channel gating.

While this finding is truly exciting, it is not sufficient to support this strong mechanistic conclusion. A number of critical experiments and conceptual developments are absolutely required to support the conclusion of a tether mechanism:

1) The authors interpretation is entirely depending on the idea that force transmission in pressure-clamp is primarily mediated by force-from-lipids, whereas in poke there is a large recruitment of long-range channels via force-from-filament. However, this framework is far from established and complicated by a number of factors:

We agree with the reviewer in that the framework is far from being established. The reviewer says that our interpretation is entirely depending on this framework. While this is one way of putting it, we believe that it is the other way around – that is our data supports this framework. We have performed several additional experiments and have significantly revised the result section and most importantly the discussion of the manuscript to emphasize this point. In the revised version we also discuss alternative concepts of channel gating and put them in the context of our findings. We hope the reviewer enjoys reading the revised paper and appreciates the changes we have made.

a) Contrary to what the authors claim, it has been previously demonstrated (Cox 2016) and is also evident from the manuscript's own data (Figure 5e-g) that the cell-attached configuration, and even inside-out patches, are not devoid of cytoskeleton or cytoskeletal influence.

We thank the reviewer for pointing out that inside-out patches (excised patches) are not devoid of cytoskeleton as we had written on page 13 line 2 of the original manuscript. We have corrected this statement in the revised version of the manuscript.

We would, however, like to point out that we never claimed that patches in the cell-attached configuration are also devoid of cytoskeleton as stated by the reviewer. We originally wrote on page 13 (line 3) "... *partially detached from the membrane (cell-attached patches)*". We are aware that many researchers believe that the cytoskeleton is largely unperturbed in cell-attached patches, but this has actually never been directly shown. We would like to note that initial patch formation occurs by blebbing into the patch pipette that disrupts normal membrane structure and detaches the membrane from the underlying actin cortex. Once the giga-seal is formed the membrane patch keeps creeping up the patch pipette driven by electroosmotic forces – a phenomenon known as seal creep. Although it is true that the cytoskeleton also protrudes into the patch-pipette (see new figure 5f), it has never been shown that the 3D molecular architecture of the membrane-cytoskeleton interface inside the patch pipette is indeed identical to that of intact cells.

We have nevertheless changed the wording regarding this issue throughout the revised version of the manuscript and we also discuss the influence of the cytoskeleton in stretch sensitivity in cell-attached patches, which, if anything, is inhibitory. We hope the reviewer appreciates the changes that were made to manuscript and considers this issue as being resolved.

b) The stretch sensitivity experiments are difficult to evaluate due to the inconsistent current responses and the fact that the stretch response of Piezo2 does not saturate prior to membrane lysis.

We agree with the reviewer. We were fully aware of this problem and had, in fact, described it in great detail in the original submission (page 6, line 17-25). We would like to emphasize that the fact that stretch-activated PIEZO2 currents do not saturate is not related to a problem in our recordings (note that our PIEZO1 currents look absolutely normal, new Supplementary Fig. 5f). In fact, it is well-known that PIEZO2 does not behave like PIEZO1 in cell-attached recordings and others have described similar PIEZO2 currents as we do.

We do, however, disagree with the reviewer in that it is difficult to evaluate the experiments. Measuring the area under the curve over the time of the pressure stimulus provides a precise measure for the total charge transfer and is just as accurate and meaningful as measuring peak current amplitudes – like most people do for PIEZO1 currents – which completely ignores channel activity after the peak.

c) The authors state the differences seen in whole-cell poke are due to the loss of long-range activation via force-from-filament, however no such long-range activation was ever shown. Even taken at face value, as demonstrated in Shi 2018, tension should diffuse over a wider area upon cytoskeletal disruption in addition to the potential of any lost long-range contacts complicating this interpretation.

In order to address all of these points the authors have to demonstrate, using an imaging or equivalent approach, that long-range activation of Piezo channels, and subsequent loss upon CytoD treatment IDR5 deletion, is actually occurring.

We agree that long-range activation of PIEZOs has not been demonstrated yet. However, considering that whole-cell PIEZO2 currents are probably mediated by more than 1000 channels (whole-cell amplitude/single channel amplitude) together with the PIEZO2 cluster density and assuming that each cluster contains ~50 channels (just a guess based on previous super-resolution studies of PIEZO1 cluster), the whole-cell currents are probably generated by channels that are distributed over a cell surface area of $60 \mu\text{m}^2$ (a circular area with a radius of $\sim 4.4 \mu\text{m}$). Hence, long-range activation of PIEZO2 (we believe that $4.4 \mu\text{m}$ can be considered “long-range” at the cellular scale) is highly likely to contribute to whole-cell poking-evoked currents.

As suggested by the reviewer we nevertheless tried to directly demonstrate the existence of long-range activation. We first explored the possibility to visualize Ca^{2+} influx through PIEZO2 channels by TIRF microscopy. To this end we performed live-cell imaging cell-substrate interface (80 Hz sampling frequency, 100x oil immersion objective) of Cal-520-AM loaded PIEZO2-expressing N2a cells while stimulating the top side of the cell. Unfortunately, we could only detect stimulation artifacts but not PIEZO2-mediated calcium influx (see figure below). This negative result does, however, not prove that long-range activation does not exist. It is equally likely, that Ca^{2+} influx through PIEZO2 cannot be detected considering the low calcium permeability of PIEZO2 and the temporally (fast inactivation of PIEZO2) as well as spatially (small cluster size) limited nature of a possible PIEZO2 response.

Figure 1. Live TIRF imaging of calcium signals from PIEZO2mScarlet- or mCherry-transfected N2A cells and stimulated with a series of mechanical stimuli with increasing amplitude ($1\mu\text{m}$ to $8\mu\text{m}$) (left). Cells were prepared for TIRF microscopy as described in the revised manuscript. For calcium imaging, cells were incubated for 30min with Cal-520 AM ($2\mu\text{M}$, AAT Bioquest) and Pluronic F-127 (0.04% final concentration). After 3 washes with the

imaging buffer (4mM KCl, 140mM NaCl, 8mM glucose, 10mM HEPES, 1mM MgCl₂, 3mM CaCl₂, pH 7.4), cells were allowed to rest for 10min before the start of the imaging (RT) The representative images and normalized changes in fluorescent traces over the entire cell (right) are representative of 30 PIEZO2-transfected cells recordings and 12 for the vector.

We thus took an alternative approach to directly demonstrate long-range activation of PIEZO2 (see new Fig. 5f). To this end we performed cell-attached patch clamp recordings and mechanically stimulated the cell surface adjacent to the patch pipette (see new Fig. 5f). In these recordings, we observed mechanically-evoked inward currents in one from 33 recorded cells (Fig. f). Although the fact that only a single cell responded in this assay was disappointing at first glance, it nevertheless showed that long-range activation in principle exists. As evidenced by lifeAct fluorescence inside the patch pipette (new Fig. 5f), the cytoskeleton is present inside the cell-attached patches, but the membrane creeps up the patch-pipette by several micrometers, which causes structural remodeling of the cytoskeleton and possibly partial detachment from the membrane, which we believe probably precludes long-range force-from-filament activation of channels inside the patch pipette in most recordings.

2) There are well-established approaches to determining a direct cytoskeletal interaction that should be performed in order to prove a tether mechanism: If IDR5 does indeed interact with a tethering protein the authors should be able to:

- a) Identify the tethering protein.
- b) Show in pull-down experiments that the tethering protein and Piezo2 interact.
- c) Perform competition assays to validate the functional interaction.

We agree that pull-down experiments and competition assays are “well-established” techniques as the reviewer says, but insinuating that the identification of the tether protein is something that could be achieved as part a paper revision goes a bit too far. Identifying a putative tether protein is anything else but easy and would be way beyond the scope of this study. Reviewer #2 also acknowledges this and explicitly states that: “*Recommending such experiments would far exceed what would be permissible in a revision*”. It took several years to understand the exact nature of tethering of the NOMPC or the TMC1 channel. Likewise, first evidence for the requirement of an extracellular tether in mouse sensory neurons was provided in 2011 (Chiang LY et al. 2011, Nat Neurosci), but we have only recently identified a candidate for this tether protein (Schwaller et al. 2021, Nat Neurosci). Moreover, it is completely unclear if there even is a tether, because PIEZO2 might as well be directly linked to the cytoskeleton. We also believe that our manuscript provides enough novel mechanistic insights and represents a major conceptual advance in the field to appeal to the broad readership of Nature Communications even without having identified the tether molecule. We thus hope that the reviewer does not insist on the identification of the tether molecule.

Minor

concerns:

1) The analyses in figure 3 should be limited to patches that show distinct single-channel openings. Geng et al. observed multiple subconductance states upon substitution of the residue corresponding to Y1568 immediately preceding IDR3. It is possible that the removal of this large proximal domain would have similar effects. Performing the analyses on patches showing overlapping channel openings prevents the identification of these states and they may spuriously combine to suggest full single-channel openings.

In fact, the analysis of single channel conductance was limited to patches that showed clearly discernable single channel opening as the ones shown in the original figure 3b and the new figures 2d and 3d. This is why the N-numbers of the single channel data are smaller than the overall N-numbers (compare original Fig. 2 with Fig. 3). We didn't explicitly mention this in the original submission, because we thought it was clear that one cannot analyze single channel conductances from traces like the ones shown in the original figure 2a. An additional statement regarding single channel analysis has now been added to the Material and Methods section.

Regarding the sub-conductance states mentioned by the reviewer, we would like to emphasize that we are of course aware of the role of Y1568. In fact, we had described the importance of this amino acid as well as the lateral plug before Geng et al. did (Taberner et al. PNAS 2019) and we had

come to the very same conclusion as the reviewer in the original version of the manuscript. We explicitly wrote: “*Considering that IDR3 spans a distance of approximately 7 nm, we hypothesize that deletion of IDR3 causes dislocation of the adjacent lateral plug, such that the lateral portals are unblocked and single channel conductance is increased.*” (page 7, line 9-12; original submission)

Inspired by the comments of reviewer #3 we have now directly tested this hypothesis and generated an additional PIEZO2 mutant in which IDR3 as well as the lateral plug are missing. We show that there is no additional increase in single channel conductance, which strengthens our original claim, which is also proposed by the reviewer, that deletion of IDR3 causes an increase in single channel conductance by affecting the binding of the lateral plug to the lateral portals.

2) It is inappropriate to refer to the intracellular domains as “disordered regions”. The term disordered makes specific inferences about the structure of these regions. Their absence in available structures implies that they are likely to be flexible, but perhaps not necessarily disordered. A nice example of why this nomenclature is not appropriate is seen in TRPA1 channels, where the ankyrin repeat domain is difficult to resolve, but extremely likely to be highly structured.

We disagree with the reviewer in that it is inappropriate to term the intracellular domains “disordered regions”. We did not choose the term IDR because the structures are unresolved, as implied by the reviewer, but because we had previously analyzed the AA sequence of PIEZO2 with respect to the presence of intrinsically disordered domains (Fig.1a in Taberner et al. 2019, PNAS) using the IUPred algorithm for the prediction of intrinsic disorder. We agree that we should have made this point clearer in the original submission of this manuscript. We have thus added a new figure to the revised version of the manuscript (new Figure 1b) in which we show the disorder probability calculated by three different disorder prediction algorithms (IUPred-L, VSL2b and ESpritz-N). All three algorithms indicate that the domains that we had originally termed ‘IDRs’ are indeed highly likely to be intrinsically disordered.

By contrast, all three algorithms show that the ankyrin repeats of TRPA1 are NOT disordered (data not shown), which is consistent with the reviewer’s comment and nicely demonstrates that the prediction algorithms are reliable and meaningful. We have added a statement at the beginning of the result section in which we describe the analysis of the amino acid sequence in order to justify our terminology. We hope this additional statement resolves the issue.

3) Figure 1a includes structural models of the IDR regions, however it is unclear how these models were generated. This should be included in the methods and the fact that these models were generated and are not a part of the original structure should be made clearer when describing the figure.

An more detailed description of how the model shown in figure 1 was generated, has been added to the figure legend of the revised manuscript.

4) (pg 12 line 16) the authors write: “This is very interesting, because a part of IDR4 is encoded by exon18, which is missing in Piezo2 splice variants that are expressed in the lung and bladder where Piezo2 is mostly exposed to stretch rather than focal indentation of the cell membrane. Hence, a role of IDR4 in fine-tuning Piezo2 stretch-sensitivity would make perfect sense.” However, this line of reasoning is not well supported. Even in the skin the majority of stimuli will deform the tissue on a scale much larger than the size of a cell resulting in stretch as opposed to a focal indentation. The authors should tone down or remove this statement.

We have toned down this statement in the revised manuscript as suggested by the reviewer.

We would nevertheless like to point out that cells in the lung and bladder very likely experience different types of mechanical stimuli than cutaneous sensory afferents. Thus, in the bladder and the lung, PIEZO2 is present at the surface of the cell bodies which are stretched as a whole during bladder distention and lung inflation, respectively. By contrast, the nerve endings of cutaneous sensory afferents are associated with highly specialized end-organs such as hair-follicles, Meissner corpuscles, Pacinian corpuscles and Merkel cells, which supposedly bundle or focus mechanical stimuli onto rather small spots on the sensory afferent endings. Li & Ginty, for example, have

described thin filamentous connections between hair follicle epithelial cells and sensory afferent endings, which might serve as extracellular tethers (Li & Ginty, *elife* 2014). Likewise, we have recently shown that USH2A might serve as an extracellular tether in Meissner corpuscles (Schwaller et al. *Nat Neurosci.* 2021). Hence, in contrast to lung and bladder cells, cutaneous afferents are very likely to experience very focal mechanical stimuli.

Moreover, it is simply not true that “...in the skin the majority of stimuli will deform the tissue on a scale much larger than the size of a cell” as the reviewer claims. The physiologically relevant stimuli that activate cutaneous afferent are much smaller. I have personally measured the mechanical activation thresholds of cutaneous low-threshold mechanoreceptors – i.e. sensory afferents that detect tactile stimuli – using human psychophysics and skin-nerve recordings in mice (Lechner & Lewin, *J Physiol* 2009; Wende et al. *Science* 2012; Heidenreich et al. *Nat Neurosci* 2012; Moshourab et al. *JOVE* 2016; Arcourt et al. *Neuron* 2017; Dhandapani et al. *Nat Commun.* 2018; Schwaller et al. *Nat. Neurosci.* 2021) and I can assure the reviewer that the thresholds are in the range of 20 μm and below in both species. Note, this is the activation threshold at the skin surface, which means that the actual displacement at the nerve ending, where PIEZO2 is located, is even smaller.

Reviewer #2 (Remarks to the Author):

An intrinsically disordered intracellular domain of PIEZO2 is required for force-from-filament activation of the channel

Verkest...Lechner *Nature Communications*

In the present manuscript, the authors aim to tackle PIEZO structure-function. This is an interesting area of study, particularly because the authors focus on structure-function of PIEZO2, which is less understood than PIEZO1. Given that splice variants of PIEZO2 are differentially expressed in known mechanosensitive tissues (bladder, lung, sensory neurons), and that PIEZO2 missense mutations can have a broad range of consequences for humans, it is crucial to uncover how structural domains of PIEZO2 govern mechanical gating in response to different kinds of forces. The electrophysiology data provide a detailed study illuminating the role of intrinsically disordered regions of mouse PIEZO2. This dataset will substantially advance our knowledge of PIEZO2 structure-function. Additionally, the data bring up the important point that poke and stretch, the two assays mainly used to study mechanically activated ion channels, have differing effects on PIEZO2 deletion mutants lacking specific IDRs.

We thank the reviewer for his/her enthusiasm about our work.

However, the authors go too far in their conclusions. A glaring inconsistency in the paper is the bold claim in the title and abstract that PIEZO2 is directly activated by cytoskeletal forces, with comparatively little data to back up such a strong claim. While their hypothesis is an interesting one, the IDR deletion studies do not adequately address it. To convincingly show force-from-filament, the authors would need to either specifically identify the tether molecule and salient PIEZO2 residues or demonstrate the interaction physically, as was nicely shown for NOMPC and its ankyrin repeats (Zhang et al 2015, *Cell*). Recommending such experiments would far exceed what would be permissible in a revision. We also note the recent study from Romero et al. on margaric acid dependent modulation of PIEZO2 (*Nature Communications*, 2020). They show that cytoskeletal disruption (with Latrunculin A) modulates PIEZO2 ion channel activity. However, they temper their conclusions by stating that this is evidence for “cytoskeletal regulation”. While the authors of this manuscript examine the IDRs as opposed to the beam, the level of evidence for cytoskeletal interaction of PIEZO2 is comparable. The title, abstract, and conclusions should be rephrased to support the data contained within. We have not combed through line-by-line as these changes will require a substantial rewrite of the manuscript text. These changes are essential for revision of the manuscript.

We thank the reviewer for his/her constructive comments and for his/her understanding that the identification of the putative tether protein would be way beyond the scope of this study. We also appreciate the reviewers concerns regarding our conclusion and have thus revised the manuscript

and tempered our conclusions according to the reviewer's suggestions. We hope that the reviewer agrees to the revised version.

Regarding the comparison of our study with the excellent work of Romero et al (2020) we would, nevertheless, like to point out that Romero and colleagues (2020) actually only show that disruption of the cytoskeleton by Latrunculin-A increases the sensitivity of PIEZO2 to margaric acid and that this sensitivity is controlled by the beam domain. From the published data it is, however, impossible to evaluate to what extent the beam-swap experiment or the Latrunculin treatment also affected PIEZO2 gating, because Romero et al. only show normalized data and not absolute mechanical thresholds or current amplitudes from PIEZO2 recordings. Hence, it makes sense that they interpret their results as evidence for "cytoskeletal regulation".

What I believe the authors do succeed in demonstrating is that the intrinsically disordered domains IDR3, 4, and 5 have different effects on poke versus stretch evoked currents. The findings suggest that there may be distinct functional domains of PIEZO2 that are attuned to specific mechanical stimuli and/or that there may be interacting proteins that facilitate these distinct modes of activation, cytoskeletal or otherwise. Any speculation as to the underlying mechanism should be presented as such, as the evidence is insufficient to demonstrate one way or the other. My issue with the authors' conclusions is not merely a question of semantics: a crucial flaw in the logic of the authors is their underlying assumption that because there is a poke vs. stretch difference, that it must be due to cytoskeleton vs. bilayer activation.

We fully agree with the reviewer and would like to emphasize that we have actually never claimed that poke is due to cytoskeleton and stretch is due to bilayer as stated by the reviewer. In fact, we explicitly wrote: *"Thus, we propose that pressure-induced membrane stretch in cell-attached recordings predominantly activates PIEZO2 via force-from-lipids, whereas mechanical indentation of the cell activates some channels via force-from-lipids and others via force-from-filament."* (page 14 line 21-24, original submission). We have, nevertheless, made this point much clearer in the revised manuscript and hope that this resolves this issue.

Figures 4 and 5 serve as "roundabout" experiments that do not have the potential to disprove the hypothesis. Rather, Figure 4 and 5 implicate a role for IDR5 allosteric modulation of PIEZO2 activity by PKA and affirm that IDR5 modulates modulate the mechanosensitivity of PIEZO2. Figure 6 is confusing since it is unclear as to what are the precise mechanical underpinnings of neurite outgrowth in the NGF + serum deprivation assay in the N2A cells. However, that they see an effect of PIEZO2 transfection suggests that there is indeed a mechanical component to the process worthy of future investigation, and this should be discussed. What would be more convincing than Figures 4, 5, and 6 are PIEZO1 chimera experiments with the IDRs followed by poke and stretch (where the protein homology renders this feasible).

As suggested by the reviewer, we have generated PIEZO chimeras (see detailed response to point 4 below).

We agree that the original Figure 4 was kind of a "roundabout" as the reviewer has called it and thus we excluded this figure from the revised version of the manuscript.

We, however, disagree that Fig.5 was also a "roundabout" experiment and that Fig.6 was "confusing" as the reviewer wrote.

The original figure 5 and even more so the revised figure 5, provide an absolutely essential information, that is that IDR5^{del} currents are not inhibited by disruption of the cytoskeleton. This is a key finding as it demonstrates that IDR5^{del} is activated by a mechanism that is independent of the cytoskeleton.

Figure 6 also presents an important finding, that is that an assay that depends on the detection of naturally occurring mechanical stimuli (neurite outgrowth) can reveal functional deficits of a channel that appears to function normal when examined with the pressure clamp technique (note, IDR5^{del} is indistinguishable from PIEZO2 in these recordings – see new Fig. 2). We have emphasized this point in the discussion of the revised manuscript.

Irrespective of the flawed conclusions, this paper presents important structure-function data that will advance the mechanotransduction field, even if the wording is toned down substantially, as we recommend. Below I list specific experimental concerns and suggest some new experiments

to partially address the conclusions as presented in the present manuscript.

Major Experimental concerns:

1. Mock transfected or untransfected controls must be shown for all electrophysiology experiments (poke, stretch, and single channel stretch), with comparable N. We realize that poke was shown in Moroni et al 2018, but the authors should include these for all the major electrophysiology panels in the first half of the paper (1d, 2b-c, 3b) to ensure rigor.

We have performed additional recordings in both patch-clamp configurations (poking and stretch) from N2A cells transfected with a GFP expressing vector. 15 (poking, Figure 1d) and 18 (stretch, Figure 2b-c) cells were recorded and did not show any mechano-activated current.

2. It was surprising to see that PIEZO2 could tolerate these IDR deletions and still retain function. In Figure 3G, surface biotinylation kits are notorious for labeling intracellular protein as well as extracellular. It would be better to use a more specific form of surface labeling, such as by engineering the tag onto an extracellular region and performing antibody staining in permeabilized vs. nonpermeabilized cells (Coste et al 2015 Nat Comm). Thus the claim that there is no trafficking defect is unfounded. The small changes in whole cell current for IDR5 could be explained equally well by impaired trafficking. Were this the case, it would entirely refute the mechanism proposed in the paper. This claim should be toned down or additional supporting data should be presented. While the IDR3 region clearly affects the pore, the mechanism is less clear for IDR5 and the paper hinges on there being no trafficking defect.

We agree with the reviewer's concern regarding the importance of ruling out a possible trafficking defect of IDR5^{del}. We would, however, like to point out that our original conclusion about there being no trafficking defect, did not exclusively rely on the biotinylation assay. We had also analyzed confocal images of cells in which PIEZO2 and IDR5^{del} were labelled with an anti-HA antibody in which the membrane was visualized with a fluorescent marker (original Fig 3f). Moreover, we had shown that the proportion of cells that responded to negative pressure in cell-attached recordings, was similar for PIEZO2 and IDR5^{del}-expressing cells (original Fig. 2), which also indicated that there is a similar number of functional channels in the membrane.

Inspired by the reviewers comment, we nevertheless performed additional experiments to further support our claim and to rule out a possible trafficking defect of IDR5^{del}.

While surface labelling of nonpermeabilized cells expressing a PIEZO2 construct bearing an extracellular tag, as suggested by the reviewer, would be a good option, this approach often lacks quantitative information that in our case is important.

We thus generated PIEZO2 and IDR5^{del} versions that are tagged with the red fluorescent protein mScarlet on the C-terminus and conducted a detailed single-particle tracking TIRF analysis (See Figure 4 and Supplementary Figure 6 in revised manuscript). We have determined the mean number of PIEZO2 clusters on randomly selected cells (Figure 4b) and observed even a slight tendency towards a higher number of IDR5^{del} cluster at the plasma membrane. Furthermore, we approximated the size for each cluster with a 2D Gaussian fit (Figure 4c) and did not find any size difference between PIEZO2 and IDR5^{del}. We also analyzed the diffusion properties of the PIEZO2 and IDR5^{del} clusters (Figure 4d-f and Supplementary Figure 6k) and did not find any major differences. Hence, together with our previous experiments, the new data convincingly demonstrates that IDR5^{del} is expressed at the plasma membrane to the same extent as PIEZO2.

3. A concern of the cytochalasin D experiments is that by altering the cytoskeleton, one also indirectly alters the surface tension of the cell membrane, thus affecting poke and stretch sensitivity independently of proposed PIEZO2-cytoskeleton tethers. We like to assume in these experiments that we are only altering the cytoskeleton, but for this paper especially it is crucial to affirm whether that assumption holds. It would be prudent to quantify cell volumes and surface areas to ensure the authors are accounting for any changes in cell surface tension that could indirectly link the cytochalasin D effects to IDR5 and PIEZO2 function. The authors should also discuss the role of cytoskeleton in cell attached vs. excised patch: is there no role of cytoskeleton in either? How much does cytoskeleton contribute to activation in cell attached stretch?

We fully agree with the reviewer regarding the link between the actin cortex and plasma membrane tension and that the two components dynamically influence each other. We do, however, not think

that measuring cell volumes or surface areas is sensitive enough to detect subtle changes in membrane tension in membrane subdomain and thus even a negative result (i.e. no detectable change in volume or surface) would not allow a definitive conclusion about a possible contribution of changes in membrane tension to the observed effect of Cyto-D treatment on poking-evoked PIEZO2 currents. Nevertheless, we considered this important issue in the discussion of the revised manuscript (page 10 line 7-9). Moreover, the revised discussion also considers the second point raised by the reviewer, which is the effect of the cytoskeleton on channel activation in cell-attached recording. We hope that the changes made to the discussion are sufficient to resolve this issue.

Suggestions for new experiments:

4. I would suggest PIEZO1-PIEZO2 chimera experiments (IDR swaps), similar to what was performed for the beam in Romero et al (see above). This would render the cytochalasin D experiments more convincing than a simple deletion.

We have performed the suggested IDRs swap experiment for IDR5, IDR4 and IDR2 by introducing them into PIEZO1. We found that IDR2 controls the velocity sensitivity of PIEZO2 and demonstrate that insertion of P2-IDR2 into PIEZO1 confers the same velocity sensitivity to PIEZO1 (see new Fig. 3a and b). Insertion of P2-IDR4 and P2-IDR5 into the corresponding positions in PIEZO1 did, however, not alter the poking or stretch-sensitivity of PIEZO1 (data not shown). IDR5 appears to control poking sensitivity of PIEZO2. Since full-length PIEZO1 is, however, anyway very sensitive to poking, it is not surprising that we didn't see any differences after insertion of P2-IDR5 into PIEZO1.

5. And with regards to the deletion, typically we see a glycine linker added in these experiments rather than a deletion (see Wang et al 2019 Nature). It might be worthwhile since the constructs are already made, to perform a site directed insertion of GGG or GGGG into IDRs 3,4, and 5 and re-test the construct in poke and stretch.

The different IDR^{del} mutants we had generated still contain several amino acids between the THU and the neighboring intracellular pre-transmembrane helices (see original and new Supplementary Fig. 1). Hence, we originally didn't think that it was necessary to add glycine stretches as a control. In case of IDR3^{del}, it is questionable if this would make any difference at all, because the gap that is created by deletion of IDR3 is app. 7nm (distance between N- and C-term of IDR3) wide and would thus anyway not be filled by a stretch of 3 or 4 glycines.

In case of IDR5^{del}, however, we understand the reviewers concern and have addressed this issue in the revised manuscript.

IDR5, comprises a large proportion of negatively charged AAs (33/60, new Fig. 4h and new Supplementary Fig. 1), which prompted us to hypothesize that the negative charges might be essential for the function of IDR5. We thus generated PIEZO2 mutants in which short stretches of negatively charged AAs in IDR5 were substituted by uncharged poly-alanine stretches (new Fig. 4h, PolyA1, PolyA3, PolyA4; PolyA2 cloning failed despite several attempts). While the currents mediated by PolyA3 and PolyA4 were indistinguishable from full-length PIEZO2 currents, PolyA1-mediated currents were significantly smaller and resembled those of IDR5^{del} with respect to amplitudes and mechanical activation thresholds (new Fig. 4i-l). Thus, the observed effect of IDR5 deletion is unlikely to result from a folding defect.

Reviewer #3 (Remarks to the Author):

The manuscript by Verkest et al. examines the effect of intrinsically-disordered regions (IDRs) on Piezo2 channel function. Through a careful structure-function study, the authors identify interesting new roles for some IDR domains of the protein that shed light on gating mechanisms as well as modulation of channel function. The work is significant, timely and well executed, and will be of interest to researchers in the fields of mechanotransduction and ion channel biophysics. Several points need to be addressed before publication.

Major points:

The conclusion that Piezo2-mediated inhibition of neurite outgrowth relies on the detection of cell-generated traction forces by the channel is not substantiated by experimental findings. As such,

more experiments are necessary to demonstrate that cellular traction forces activate Piezo2 through the force-from-filament mechanism mediated primarily by IDR5. Do Neuro2a cells with NGF and w/o serum show Piezo2-mediated Ca²⁺ signals on hard substrates, which are reduced on soft substrates? Are those Ca²⁺ signals abrogated by traction force inhibitors in full-length Piezo2 (e.g. Blebbistatin or ML-7) and reduced or absent in the IDR5del mutant channel? Is neurite outgrowth different on soft substrates or on hard substrates with traction force inhibitors?

The additional experiments suggested by the reviewer are all excellent and would definitely help to better understand if and how PIEZO2 detects traction forces during neurite outgrowth. However, the main focus of our study was to understand the role of the intrinsically disordered domains for PIEZO2 function. Understanding traction force detection mechanisms in N2a cells would be a study in its own right and would go beyond the scope of this revision.

The main goal of the neurite outgrowth experiment was to demonstrate that patch-clamp assays alone are not sufficient to reveal the biological relevance of a PIEZO channel mutation. We believe that the results of the neurite outgrowth assay, irrespective of whether the forces that activate PIEZO2 in this assay are traction forces or if they are of another origin, clearly make this point. If IDR5^{del} had only been examined using the pressure clamp technique, then one would have concluded that it is fully functional. Likewise, the results from the poking assay, where IDR5^{del} currents are significantly reduced but not completely abolished, do not allow definitive conclusions about the relevance of this deletion for PIEZO2 function in intact cells. It is only the combination of the three assays (pressure-clamp, poking-assays and neurite outgrowth) together with the expression level analysis that allows a meaningful interpretation of the possible biological relevance of IDR5.

We agree that it is unclear whether PIEZO2 is activated by traction forces or by other mechanical forces that occur during neurite outgrowth and we have thus rewritten the results and the discussion accordingly. We hope that these changes resolve this issue.

To satisfy the reviewers curiosity, we have nevertheless tried to record calcium signals in the neurites of PIEZO2-expressing cells during neurite outgrowth. To this end, we performed calcium imaging in TIRF mode on N2A cells 24h-40h after neurite outgrowth induction. In these experiments, we observed spontaneous calcium signals in the neurites and the cell bodies of both PIEZO2 and vector transfected cells. There was, however, no temporal correlation between the calcium signals and neurite outgrowth (e.g. change in growth direction or acceleration of growth, etc.). Moreover, the frequencies and amplitudes of the calcium signals did not differ between PIEZO2 and control vector cells (see Figure 2 below). Also, incubating the cells with the unspecific PIEZO channel blocker GsmTx4 did not affect the Ca²⁺ signals, whereas substituting extracellular calcium by barium completely abolished the signals, suggesting that the Ca²⁺ signals do not originate from PIEZO2, but probably from another ion channel.

It is important to note that PIEZO2 is less permeable to calcium than PIEZO1. Hence, it is very likely, that Ca²⁺ influx through PIEZO2 cannot be detected considering its low calcium permeability and the temporally (fast inactivation of PIEZO2) as well as spatially (small cluster size) limited nature of a possible PIEZO2 response. The uncertainty about whether or not PIEZO2 Ca²⁺ signal can be detected at all, together with the fact that there was no temporal correlation between the observed Ca²⁺ signals and neurite outgrowth, makes it impossible to interpret the results of these preliminary experiments and hence we prefer not to show them in the revised version of the manuscript.

Figure 2. Live TIRF imaging of spontaneous calcium signals from PIEZO2 and vector transfected N2A cells, serum-starved and treated with NGF for 24-40 hours. Cells were prepared for TIRF microscopy as described in our revised manuscript. For calcium imaging, cells were incubated for 30min with Cal-520 AM (2 μ M, AAT Bioquest) and Pluronic F-127 (0.04% final concentration). After 3 washes with the imaging buffer (4mM KCl, 140mM NaCl, 8mM glucose, 10mM HEPES, 1mM MgCl₂, 3mM CaCl₂, pH 7.4), cells were allowed to rest for 10min before the start of the imaging (room temperature). Images were acquired with a 100x oil objective at 20-25Hz sampling rate for 30s, with a 488nm laser excitation. The representative images (top), at 3 different time points (black arrows) and normalized fluorescent traces (bottom) over the 3 various regions (green) are representative of 31 (PIEZO2) and 30 (vector) cells.

Figure 3. Analysis of the mean frequency per cell (left) and the mean amplitude per cell (right) of spontaneous calcium signals during neurite outgrowth in PIEZO2 and vector transfected cells in control solution and with barium or GsMTx4. Detection and quantification of calcium events was performed with the ImageJ plugin xySpark.

The finding regarding the greater stretch sensitivity of IDR4^{del} is very interesting. Can the authors comment on the possible underlying mechanisms?

We also thought that this is an interesting finding especially considering that IDR4 is only half as long in PIEZO1. Hence, we generated a PIEZO1-P2IDR4 chimera in order to test if the insertion of PIEZO2-IDR4 into PIEZO1 renders PIEZO1 less sensitive to membrane stretch. This was, however, not the case, which is why we didn't follow up on this finding. Since reviewer#1 and reviewer#2 asked us to be way less speculative in our conclusions, we did not discuss the possible reasons for the increased stretch sensitivity of IDR4^{del} in the revised manuscript, as this would indeed be pure speculation not being backed up by experimental data. We hope the reviewer understands.

Figure 3: The proposed mechanism for increase in single channel conductance with the IDR3^{del} mutation is intriguing and would be strengthened by an experimental confirmation, for instance with a double deletion of the lateral plug.

This is indeed an excellent idea. As suggested by the reviewer, we generated a double deletion in which IDR3 and the lateral plug were deleted. The double deletion is functional and single channel analysis revealed no difference in single channel conductance between IDR3^{del} and the IDR3+lateralPlug^{del} (new Figure 3c and d), which strengthens our original hypothesis that the lateral plug is "unplugged" by deletion of IDR3.

Figure 4, Page 8: "our data suggest that the PKA dependent potentiation of PIEZO2 observed in our expression conditions does not require phosphorylation of the channel and is probably mediated by an indirect mechanism." While this is a possibility, it is also likely that the deletion of IDR5 affects phosphorylation of other sites. Thus, this claim should either be substantiated with direct biochemical evidence or significantly tempered.

As suggested by one of the other two reviewers, we have completely removed the PKA figure and all related text from the revised manuscript.

Figure 5: Traces should also be shown for Piezo1 stretch activation of IDR5^{del} with cytoD and Nocodazole (Fig. 5e). Also, does Nocodazole have an effect on IDR5^{del} channels in poking mode (Fig. 5h)? It may also be helpful to examine the effect of an actin stabilizer, e.g. jasplakinolide, on channel activity.

As suggested by the reviewer, we have performed additional experiments with Jasplakinolide and have also examined the effect of Cyto-D and Nocodazole on stretch-activated IDR5^{del} currents. The new data including the requested example traces is shown in the new Figure 5 and supplementary Figure 7.

Minor points:

Page 4, line29-30: Please summarize the evidence for intracellular tethering for NOMPC and TMC1

I am afraid we have to deny the reviewers request regarding a more detailed summary of NOMPC and TMC1 tethering. We performed numerous revision experiments that required detailed description in the text. Moreover, the other two reviewers also asked for substantial rewriting and more detailed explanation of other things. Hence, to cut a long story short, we ran out of space and simply didn't find a way to provide a detailed description of the evidence for intracellular tethering of NOMPC and TMC1. Since the mechanisms by which NOMPC and TMC1 are tethered to the cytoskeleton are, however, not related to PIEZO2, which doesn't have ankyrin repeats like the other two channels, we think it is acceptable not to discuss this in great detail and we hope that the reviewer shares our opinion.

Page 10, line 25-26: Also cite Pathak et al. PNAS 2014, 111 (45) 16148-16153, which was the first report of Piezo channel activation by cellular traction forces.

We have cited Pathak et al. in the introduction of the revised manuscript.

In Figure 1, it would be useful to have a schematic of Piezo2 showing the IDR sites, similar to that in Fig. 4A.

The schematic that was originally shown in Fig. 4a is now shown in supplementary figure 1b. We decided not to show this schematic in Figure 1 as suggested by the reviewer, because the structural model that we show instead represents the size relation between the channel and the IDRs much better than the schematic. We believe that this is an important message that should come across, because it shows how big the IDRs actually are.

Figure 2a, please also show a family of ionic current traces for different pressure levels.

The requested traces are shown the new supplementary Figure 4.

Figure 3B: Single channel traces and amplitude histogram for IDR4del also should be shown.

Additionally, the pressure used should be mentioned

As suggested, the revised figure 2 also shows single channel traces and amplitude histogram for IDR4^{del}.

For clarity, it would be helpful to make the two arrows of different colors in Fig. 3e, and to use different colors for the plug and the latch domains.

Agreed. We have changed the color of the latch and the style of one of the arrows.

REVIEWER COMMENTS

Reviewer #1 (Remarks to the Author):

Most of my concerns have been addressed and the manuscript has greatly improved. Still, a few important concerns remain, or were generated by the revision, and must be addressed. Fortunately, all is limited to text editing – no experiments are required. After these changes are completed the work will be an important contribution to the field and I am looking forward to seeing it published.

Major:

1. In their revision the authors perform experiments in an attempt to separate a direct force-from-filament model from a force-from-lipids-via-filament model (Figure 5e, and f; page 11; lines 7-26). However, the result is not at all robust or repeatable, with only 1/33 cells showing the ‘desired’ response. Still, from this single outlier the authors draw the bold conclusion that force is transmitted to the patch dome by a force-from-filament mechanism. This interpretation is clearly biased and likely wrong. In addition, other unrelated mechanisms are possible that may well explain this single outlier: For example, poke may induce a change in cell internal fluid pressure, which could stress the patch dome and thereby activate channels.

This experiment is not suited to test the outlined hypothesis, the data interpretation is massively biased, and therefore this section should be removed entirely from the manuscript.

Minor:

2. Figure 4d-f (page 15 lines 15-17) show that Piezos remain highly mobile in the absence of IDR5. This is contrary to a tether mechanism, which should render proteins immobile. The authors should discuss this contradiction in their manuscript.

3. Page 7; line 18. The authors claim that IDR3 keeps the lateral plug in place, but only structural data could ever provide proof of such a strong claim. This statement should be removed from the manuscript.

4. Page 13; lines 10-14. The authors claim that channel closure is not the mechanisms underlying the speed dependent response of LTRMs. Their own data certainly suggest an alternative mechanism, but without direct experimental testing, such a strong statement should not be made, and therefore has to be removed from the manuscript.

5. Figure 1B: The colors in the legend do not appear to match the prediction traces in the figure.

Reviewer #2 (Remarks to the Author):

The authors have made substantial revisions to the manuscript, including: the removal of confusing experiments that did not directly support the conclusions, tempering of conclusions related to force-from-lipids vs. force-from-filament, and inclusion of new data such as domain-swap experiments and added controls that bolster the findings. The manuscript text has undergone substantial revision throughout, including the abstract and title, to ensure the data support the conclusions. Speculative statements are explicitly marked as such and lay ground for future work. It is our opinion that our concerns have been adequately addressed.

We have one minor point of revision: p. 13 line 12, "IDR2" was misspelled "IRD2".

Reviewer #3 (Remarks to the Author):

The authors have included several new experiments in the revised submission. However, important issues related to the original comments remain and some new ones are generated by the new data:

- The conclusion that IDR5 is required for PIEZO2 activation by "cytoskeletal derived forces" is not supported by the data. What the authors show is that the actin cytoskeleton is required for robust activation of PIEZO2 by poking. To say that these forces are "cytoskeletal derived" is incorrect since the mechanical force is provided by the experimental poking stimulation. "Cytoskeletal derived" indicates that the forces are generated by the cellular cytoskeleton which is not the case here. A more appropriate description would be that the forces are "transmitted by the cytoskeleton". The authors should be careful about using term "cytoskeletal derived forces" elsewhere in the manuscript as well, e.g. Discussion page 14, line 21 and page 15, line 1.

- The results from the cell-attached patch experiments with poking in the vicinity are inconclusive – 1 patch out of 33 is not a robust response and it is possible that the signal in the patch that responded may have arisen from technical artifacts. Hence, these data should be removed from the paper.

- I commend the authors in attempting the experiments related to examining traction force induced activation of PIEZO2. It is unfortunate that the experiments were unsuccessful. The existing data does not support the claim PIEZO2 is activated by cell-generated forces or that IDR5 is required for the detection of cell-generated forces during neurite outgrowth. While such roles may be hinted at by the difference in neurite outgrowth assay, other explanations are also possible (e.g. an effect mediated by a ligand binding to PIEZO2, and other possibilities). A lot more work needs to be done to support the conclusions presented by the authors and to address the question posed in this section “how naturally occurring stimuli in intact cells activate PIEZO2” and “how the cell-generated forces activate the channel”. These are best explored in a separate study. As such, the neurite outgrowth data and any associated interpretation should be removed from the manuscript.

REVIEWER COMMENTS

Reviewer #1 (Remarks to the Author):

Most of my concerns have been addressed and the manuscript has greatly improved. Still, a few important concerns remain, or were generated by the revision, and must be addressed. Fortunately, all is limited to text editing – no experiments are required. After these changes are completed the work will be an important contribution to the field and I am looking forward to seeing it published.

We thank the reviewer for his/her positive and enthusiastic feedback.

Major:

1. In their revision the authors perform experiments in an attempt to separate a direct force-from-filament model from a force-from-lipids-via-filament model (Figure 5e, and f; page 11; lines 7-26). However, the result is not at all robust or repeatable, with only 1/33 cells showing the ‘desired’ response. Still, from this single outlier the authors draw the bold conclusion that force is transmitted to the patch dome by a force-from-filament mechanism. This interpretation is clearly biased and likely wrong. In addition, other unrelated mechanisms are possible that may well explain this single outlier: For example, poke may induce a change in cell internal fluid pressure, which could stress the patch dome and thereby activate channels.

This experiment is not suited to test the outlined hypothesis, the data interpretation is massively biased, and therefore this section should be removed entirely from the manuscript.

We appreciate the reviewers concern and have thus removed this data from the manuscript.

Minor:

2. Figure 4d-f (page 15 lines 15-17) show that Piezos remain highly mobile in the absence of IDR5. This is contrary to a tether mechanism, which should render proteins immobile. The authors should discuss this contradiction in their manuscript.

We agree with the reviewer that – at first glance – our TIRF results and the lack of differences in diffusion behavior of PIEZO2 and IDR5^{del} clusters, respectively, do not seem to indicate a potential tethering mechanism. We would, however, like to point out that we never claimed that IDR5 directly tethers PIEZO2 to the cytoskeleton. The words tether and tethered are only used three times throughout the entire manuscript, but not in the context of IDR5^{del}. It is of course possible that IDR5 is solely required for the activation of PIEZO2 via the cytoskeleton, while another channel domain or an auxiliary subunit is required for the actual physical tethering. In fact, in the last sentence of the discussion, we actually explicitly stated that “*An important question that remains open and calls for further investigation is whether IDR5 directly links PIEZO2 to the cytoskeleton or if a separate tether protein mediates this interaction*”.

However, the fact the reviewer got the impression that we insinuate that IDR5 is involved in tethering, shows that we should have done a better job in explaining this point. We thus added a short section to the discussion (page 15, line 1-17, highlighted in yellow) in which we discuss

possible reasons for why the diffusion behavior of IDR5^{del} does not differ from that of full-length PIEZO2. We hope the new discussion section resolves this issue.

3. Page 7; line 18. The authors claim that IDR3 keeps the lateral plug in place, but only structural data could ever provide proof of such a strong claim. This statement should be removed from the manuscript.

We agree that only structural data will provide ultimate proof of such a claim. Hence, as suggested by the reviewer, we have changed the wording of the subheading in the results section (page 7 line 18).

We have, however, not re-written the result section or the discussion, because we believe that both sections had already been written very carefully and had made it absolutely clear that the idea that IDR3 keeps the plug in the right position is pure speculation. In the result section we state that “...,IDR3 *appears to be an important flexible structural element that ensures proper positioning of the lateral plug.*” (highlighted in purple, page 8 line 2) and in the discussion we solely state that “...*deletion of IDR3 significantly increases single channel conductance, probably by dislocating the lateral plug from the lateral ion conducting portals*” (highlighted in purple, p12/l32 – p13/l3). We believe that the words ‘appear’ and ‘probably’ clearly indicate the speculative nature of these statements and we hope that the editor and the reviewer agree that it is sufficient to change the wording of the subheading to resolve this issue.

4. Page 13; lines 10-14. The authors claim that channel closure is not the mechanisms underlying the speed dependent response of LTRMs. Their own data certainly suggest an alternative mechanism, but without direct experimental testing, such a strong statement should not be made, and therefore has to be removed from the manuscript.

We appreciate the reviewers concern, but we really don’t know how to change the text in order to make satisfy the reviewer.

The reviewer claims that we say “*that channel closure is not the mechanism underlying the speed dependent response of LTMRs*”. This is not true! We solely state that “*our data CONTRADICTS this hypothesis*” (page 12 line 26), which the reviewer obviously agrees with as he/she states that our data “*certainly suggest an alternative mechanism*”. We believe that saying that a certain finding contradicts with a previous hypothesis is not a “strong statement” as the reviewer calls it and we really don’t understand why this statement should be removed or how we could rephrase it to satisfy the reviewer. We only put our data in the context of the existing literature, which is exactly what should be done in the discussion of paper. In fact, we have even explicitly mentioned that our data does “*NOT allow any conclusions about the mechanistic basis of this important property*” (page 12 line 31-32). Hence, we believe that we couldn’t have discussed this issue more carefully and diplomatically than we did and would therefore like to keep this statement in the manuscript in the present form. Removing this statement, as suggested by the reviewer, would mean to completely ignore the conflict between our work and the hypothesis proposed by Rugiero et al, which, in our view, would be scientifically insincere.

We would also like to point out that Rugiero et al. had only SUGGESTED that rapid channel closure is responsible for the velocity dependence of RA-currents in DRG neurons. Thus they stated that “*...the results are consistent with channels mediating RA currents closing soon after their activation so that at the end of longer ramps not all RA channels contribute to the peak current amplitude.*”. They did, however, not provide direct experimental evidence or proof of any kind that supports the idea that rapid channel closure is the main factor that controls

velocity sensitivity of PIEZO2. In fact, back at the time when Rugiero et al. carried out their experiments (before 2010), PIEZO2 has not even been cloned yet and thus Rugiero et al. had no tools to ensure that all the currents they were recording were mediated by PIEZO2.

5. Figure 1B: The colors in the legend do not appear to match the prediction traces in the figure.

We thank the reviewer for pointing this out. We have corrected the colors in the legend accordingly.

Reviewer #2 (Remarks to the Author):

The authors have made substantial revisions to the manuscript, including: the removal of confusing experiments that did not directly support the conclusions, tempering of conclusions related to force-from-lipids vs. force-from-filament, and inclusion of new data such as domain-swap experiments and added controls that bolster the findings. The manuscript text has undergone substantial revision throughout, including the abstract and title, to ensure the data support the conclusions. Speculative statements are explicitly marked as such and lay ground for future work. It is our opinion that our concerns have been adequately addressed.

We have one minor point of revision: p. 13 line 12, “IDR2” was misspelled “IRD2”.

We thank the reviewer for his/her positive feedback and for pointing out the typo on page 13, which has been corrected in the revised manuscript.

Reviewer #3 (Remarks to the Author):

The authors have included several new experiments in the revised submission. However, important issues related to the original comments remain and some new ones are generated by the new data:

- The conclusion that IDR5 is required for PIEZO2 activation by “cytoskeletal derived forces” is not supported by the data. What the authors show is that the actin cytoskeleton is required for robust activation of PIEZO2 by poking. To say that these forces are “cytoskeletal derived” is incorrect since the mechanical force is provided by the experimental poking stimulation. “Cytoskeletal derived” indicates that the forces are generated by the cellular cytoskeleton which is not the case here. A more appropriate description would be that the forces are “transmitted by the cytoskeleton”. The authors should be careful about using term “cytoskeletal derived forces” elsewhere in the manuscript as well, e.g. Discussion page 14, line 21 and page 15, line 1.

We thank the reviewer for pointing out this semantic error. We had used the term “cytoskeleton-derived forces” to distinguish these forces from cell-generated forces, so we believe that our original intention was correct but obviously not precise enough. We have thus corrected the wording throughout the manuscript as suggested by the reviewer. The changes are highlighted in yellow in the revised manuscript.

- The results from the cell-attached patch experiments with poking in the vicinity are inconclusive – 1 patch out of 33 is not a robust response and it is possible that the signal in the patch that responded may have arisen from technical artifacts. Hence, these data should be removed from the paper.

The same issue was also raised by reviewer #1 and the data has thus been removed from the revised manuscript.

- I commend the authors in attempting the experiments related to examining traction force induced activation of PIEZO2. It is unfortunate that the experiments were unsuccessful. The existing data does not support the claim PIEZO2 is activated by cell-generated forces or that IDR5 is required for the detection of cell-generated forces during neurite outgrowth. While such roles may be hinted at by the difference in neurite outgrowth assay, other explanations are also possible (e.g. an effect mediated by a ligand binding to PIEZO2, and other possibilities). A lot more work needs to be done to support the conclusions presented by the authors and to address the question posed in this section “how naturally occurring stimuli in intact cells activate PIEZO2” and “how the cell-generated forces activate the channel”. These are best explored in a separate study. As such, the neurite outgrowth data and any associated interpretation should be removed from the manuscript.

We agree that direct proof of PIEZO2 activation by cell-generated forces is missing in our manuscript. We have, however, not completely removed the neurite outgrowth data from the manuscript as suggested by the reviewer, but have moved it to the supplementary file as proposed by the editor. To address the reviewer's concern, we have toned down the wording of the conclusions drawn from these experiments (changes highlighted in yellow in the revised text; e.g. we added the possibility that PIEZO2 might detect other cues and deleted the statement about cell-generated forces from page 12 line 20-24 from the previous version). We hope that these changes satisfy the reviewer and resolve this issue.

We would like to point out that the main take-home message of the neurite outgrowth experiment is not that PIEZO2 is activated by cell-generated forces, but that channel mutants that respond normally in the stretch assay, can fail to function in a more physiological assay, regardless of whether the ‘natural’ stimulus is a cell-generated mechanical stimulus or, as suggested by the reviewer, a ligand. This is a central finding of our study as it highlights the importance of complementing patch-clamp experiments with additional assays to ensure that one does not miss a loss-of-function mutation when characterizing PIEZO channel mutants. We are glad that the editor agrees and suggested to keep the data in the supplementary file and hope that the reviewer also agrees.